# VISUAL PROMPT-AGNOSTIC EVOLUTION

**Junze Wang**[1*]**, Lei Fan**[2*]**, Dezheng Zhang**[1]**, Weipeng Jing**[3]**, Donglin Di**[4]**, Yang Song**[2]**,
Sidong Liu**[5]**, Cong Cong**[5†]

[1]University of Science and Technology Beijing, [2]University of New South Wales
[3]Northeast Forestry University, [4]Tsinghua University, [5]Macquarie University
wangjunze@yeah.net,lei.fan1@unsw.edu.au,thomas.cong@mq.edu.au

## ABSTRACT

Visual Prompt Tuning (VPT) enables effective adaptation of a frozen Vision Transformer (ViT) to downstream tasks by inserting a small number of learnable prompt tokens into the token sequence at each layer. However, we observe that existing VPT variants often suffer from unstable training dynamics, characterized by gradient oscillations. A closer layer-wise analysis reveals that shallow-layer prompts tend to stagnate early, while deeper-layer prompts exhibit high-variance oscillations, leading to a cross-layer mismatch. These issues contribute to slower convergence and degraded final performance. To address these challenges, we propose the Prompt-Agnostic Evolution (`PAE`) method, which can strengthen vision prompt tuning by explicitly modeling the dynamics of learnable prompts. From a frequency-domain perspective, we initialize prompts in a task-aware direction by uncovering and propagating frequency shortcut patterns that the backbone inherently exploits for recognition. To ensure coherent evolution across layers, we further employ a shared Koopman operator, which imposes a global linear transformation rather than uncoordinated, layer-specific updates. Finally, inspired by Lyapunov stability theory, we introduce a regularizer that constrains error amplification during evolution. Extensive experiments demonstrate that using `PAE` with VPT variants not only accelerates convergence with an average $1.41\times$ speedup but also yields 1–3% gains on 25 datasets with multi downstream tasks. Beyond performance, `PAE` remains prompt-agnostic and lightweight, and it integrates seamlessly with diverse VPT variants without backbone modification or inference-time changes, providing a practical and scalable solution for advancing prompt tuning[1].

## 1 INTRODUCTION

With the advent of large-scale vision foundation models, particularly those based on Vision Transformers (ViTs) (Dosovitskiy et al., 2020), significant progress has been made in learning general-purpose visual representations that can be transferred to a wide range of downstream tasks (Zhu et al., 2025; Fan et al., 2025a;b; Cong et al., 2025). However, effectively adapting these powerful pretrained models to specific tasks remains a key challenge, especially under constraints such as limited labeled data and computational resources. To address this, prompt tuning has emerged as a lightweight and parameter-efficient solution, enabling task-aware adaptation without modifying the backbone network (He et al., 2023; Mai et al., 2025). Among these approaches, Visual Prompt Tuning (VPT) (Jia et al., 2022) has shown particular promise for ViTs. By introducing a small set of learnable prompt tokens, VPT enables effective adaptation to new tasks while preserving generalizable knowledge encoded in the pretrained backbone (Xiao et al., 2025c).

Subsequent VPT variants improve upon this along four main directions: *Structured prompts* (Jia et al., 2022; Han et al., 2023; Tu et al., 2023; Xu et al., 2025; Ren et al., 2025; Wang et al., 2025b), which enrich input with additional channels, *e.g.*, E2VPT (Han et al., 2023) adds key/value prompts and prunes tokens for efficiency, and ProVP (Xu et al., 2025) reorganizes prior-layer outputs and applies contrastive learning to reduce mismatch; *Adaptive prompting mechanism*s (Wang et al., 2022;

---

[*]Equal contribution.
[†]Corresponding author.
[1]The code is available at https://github.com/reeive/PAE.

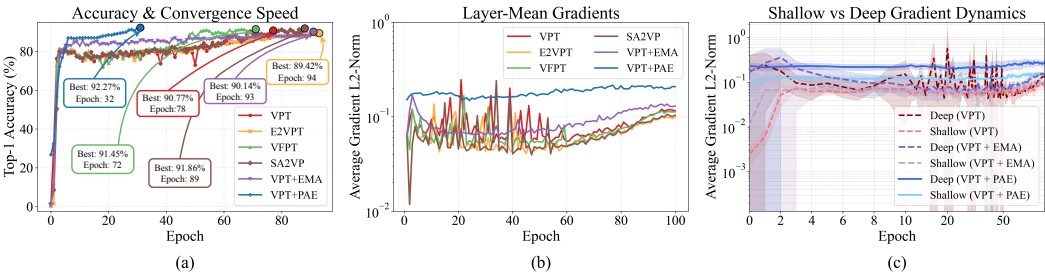

Figure 1: (a) Comparison of accuracy and convergence speed is shown in multiple VPT variants, including VPT, VPT+EMA, E2VPT, VFPT, and SA2VP. (b) Gradient oscillation (12 layers mean) is observed in multiple VPT variants, *i.e.,* VPT, E2VPT, VFPT, and SA2VP. (c) VPT hierarchically exhibits shallow-layer (Layers 1–4) stagnation and deep-layer (Layers 9–12) oscillations.

Dong et al., 2023; Sohn et al., 2023), where methods like LPT (Dong et al., 2023) dynamically combine shared and group-specific prompts for long-tailed tasks; *Projection-based prompts* (Xiao et al., 2025a;b; Liu et al., 2025) tightly couple input projection with prompt optimization, and *Perception-driven designs* (Xu et al., 2024; Zeng et al., 2024; Wang et al., 2024; Zhou et al., 2024), such as VFPT (Zeng et al., 2024), which reweights spectral components in the Fourier domain. Intuitively, since VPT and its variants update only a small number of prompt parameters, one might expect faster and more efficient training. In practice, however, they often suffer from slower convergence and suboptimal accuracy (Figure 1a). To understand this discrepancy, we analyze their gradient behaviors during training. Figure 1(b) reveals that many VPT variants exhibit pronounced gradient oscillations, particularly in the early and middle training stages. Further, our layer-wise gradient analysis in Figure 1(c) shows a clear mismatch in optimization dynamics between shallow and deep layers. Specifically, prompts inserted into shallow layers experience an early surge in gradient magnitude followed by stagnation, remaining close to their initialization, while deeper-layer prompts begin to oscillate heavily after this stagnation sets in. These phenomena suggest poor coordination across layers and contribute directly to the observed degradation in training efficiency and performance.

We attribute these phenomena to two persistent challenges in VPT variants. (1) **Task-agnostic Prompt Initialization**. While prior works have explored various initialization strategies for prompt tokens (Jia et al., 2022; Wang et al., 2024; Xu et al., 2025; Kang et al., 2025), these initializations are generally agnostic to the downstream task. As a result, early gradient updates tend to align the prompts with the pretrained backbone rather than the target task (Wang et al., 2025b). To expedite this alignment phase, many methods employ aggressive optimization setting with high initial learning rates which can exacerbate instability and induce gradient oscillations, ultimately hindering convergence and degrading performance. (2) **Independent prompt design across layers**. In most VPT variants, prompts are pre-pended and optimized independently at each transformer layer. During training, only representations propagate forward through the model, while gradients must back-propagate through many frozen layers. This leads to significantly weakened gradient signals in the shallow layers, causing their prompts to stagnate near initialization values (Merlin et al., 2023). Meanwhile, deeper-layer prompts are repeatedly adjusted in compensation, resulting in amplified gradient oscillations. The lack of explicit cross-layer coordination introduces an optimization mismatch across layers, contributing to slower convergence and reduced final accuracy.

To address Challenge (1), we propose Modal Pre-Alignment (`MPA`), inspired by recent findings that pretrained vision backbones often rely on specific frequency shortcuts to make correct prediction (Wang et al., 2025a). `MPA` performs a lightweight search to identify these task-aware shortcuts and uses them to initialize the visual prompts. To address challenge (2), we reformulate prompt optimization as a Koopman (Mezić, 2013)-Lyapunov (Shevitz & Paden, 1994) discrete dynamical system (`KLD`). In this formulation, the prompt embedding at each layer is treated as the system state, and its evolution is governed by a single shared operator that maps the prompt state from one layer to the next. Rather than optimizing each layer's prompts independently, this perspective explicitly couples them through a common dynamical rule, thereby establishing cross-layer dependencies.

Building on these two components, we introduce Prompt-Agnostic Evolution (`PAE`), a unified method designed to stabilize and accelerate prompt learning for VPT variants. `PAE` operates in two sequential stages: First, `MPA` provides task-aware prompt initialization by leveraging frequency-domain cues aligned with the backbone's inductive biases. Then, `KLD` evolves these prompts across

layers using a shared Koopman operator, enforcing smooth inter-layer transitions. A Lyapunov-style regularizer further enhances stability by constraining error accumulation during evolution. Notably, PAE is lightweight, introduces no inference-time overhead, and can be seamlessly integrated into existing methods without modifying the backbone. Our contributions are summarized as follows:

- To the best of our knowledge, this is the first work to reframe VPT as the control of prompt trajectories within a dynamical system. This offers an explicit cross-layer association perspective for VPT variants.
- We propose MPA, a task-aware prompt initialization which utilizes frequency-domain cues to produce prompts that are aligned with the downstream objective from the start.
- We propose KLD which employs a Koopman evolution operator to establish cross-layer association of prompts, while ensuring the stability of the entire optimization process by a Lyapunov-style regularizer.
- Extensive experiments on 25 datasets establish new state-of-the-art performance by incorporating PAE into various VPT variants, yielding faster convergence without incurring additional inference-time costs.

## 2 METHOD

### 2.1 PRELIMINARIES AND PROBLEM FORMULATION

**Formulation.** Given a dataset $\mathcal{D}$ consisting of input-label pairs $(x, y)$, where $x \in \mathbb{R}^{H \times W \times 3}$ is an image and $y \in \{1, \ldots, C\}$ is the corresponding class label, the image $x$ is processed by a Vision Transformer (ViT) backbone $F$, composed of $L$ encoder blocks $\{E_i\}_{i=1}^L$. Specifically, $x$ is first passed through a patch embedding layer, which splits it into $N$ patch tokens. As shown in Figure 2(a), a learnable visual prompt $\mathbf{P}_i \in \mathbb{R}^{T \times d}$ is pre-pended to the patch tokens at each transformer layer $i$, where $T$ denotes the prompt length and $d$ is the embedding dimension. We denote the set of prompts across all layers as $\mathcal{P} = \{\mathbf{P}_1, \ldots, \mathbf{P}_L\}$. These prompts are optimized while keeping the ViT backbone $F$ frozen. The final classification head $H$ is trained jointly with $\mathcal{P}$ and the overall training objective is defined as:

$$\min_{\mathcal{P}} \ \mathbb{E}_{(x,y) \sim \mathcal{D}} \ \mathcal{L}_{\text{task}}\big(H(F(x; \mathcal{P})), y\big), \tag{1}$$

where $\mathcal{L}_{\text{task}}$ is a task-aware loss function. The goal of PAE is to accelerate and stabilize the learning of prompts $\mathcal{P}$ by combining two complementary components: MPA, which provides task-aware initialization, and KLD, which enforces coordinated optimization across layers.

### 2.2 INITIALIZATION VIA MODAL PRE-ALIGNMENT (MPA)

To initialize $\{\mathbf{P}_i^{\text{init}}\}_{i=1}^L$ with task awareness, we propose the MPA strategy. As illustrated in Figure 2(b), MPA consists of two stages. First, we identify frequency shortcuts by probing $F$ using the training dataset $\mathcal{D}$. Next, we use the identified shortcuts to construct the first-layer prompt $\mathbf{P}_1^{\text{init}}$, which is then propagated through $F$ to generate the complete initialization set $\{\mathbf{P}_i^{\text{init}}\}_{i=2}^L$.

**Phase I — Discovering Frequency Shortcuts.** Neural networks often exploit the most distinctive frequency components in data as shortcuts during training (Wang et al., 2023). However, such frequency biases are not explicitly represented in the spatial domain, making them difficult to observe or control directly. To address this, we transform images into the spectral domain, where frequency characteristics are more accessible for analysis. Specifically, we begin by sampling a mini-batch $\{(x_j, y_j)\}_{j=1}^B$ from the training set. Each image $x_j$ is transformed into the frequency domain using the 2D Fourier transform, yielding $\mathcal{F}(x_j) \in \mathbb{R}^{H \times W}$. To identify task-aware regions in the frequency spectrum, we define a set of binary masks $\{\mathbf{M}_s\}_{s=1}^S$, each corresponding to a local region of the frequency space. These masks are generated by sliding a window of size $w \times w$ with stride $r$ along both height and width, resulting in $S = \left(\lfloor \frac{H-w}{r} \rfloor + 1\right) \times \left(\lfloor \frac{W-w}{r} \rfloor + 1\right)$ distinct window locations. Each mask $\mathbf{M}_s \in \{0, 1\}^{H \times W}$ has ones in its selected $w \times w$ region and zeros elsewhere. We then apply each mask $\mathbf{M}_s$ to the frequency representation $\mathcal{F}(x_j)$ by element-wise multiplication, and reconstruct the corresponding spatial-domain image using the inverse Fourier transform:

$$\hat{x}_{s,j} = \mathcal{F}^{-1}\big(\mathbf{M}_s \odot \mathcal{F}(x_j)\big), \tag{2}$$

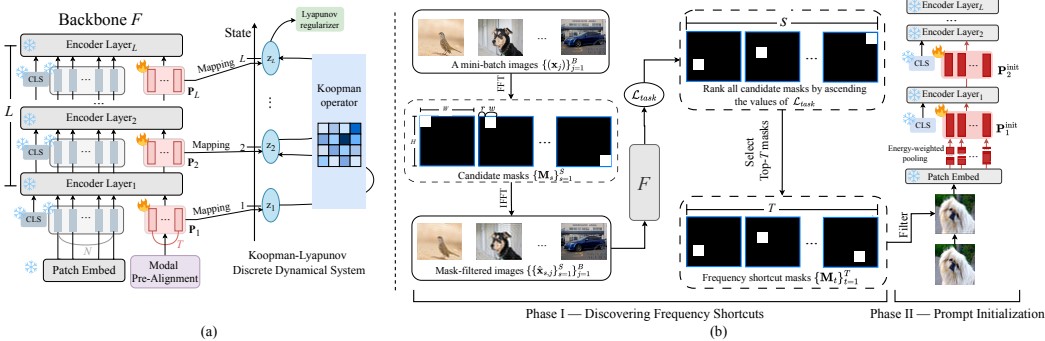

Figure 2: (a) `PAE` pipeline: `MPA` first initialize per-layer prompts. `KLD` then propagates prompts across layers via a shared Koopman operator, with a Lyapunov-style regularizer constraining error growth. (b) `MPA` pipeline: Frequency-domain transformations generate candidate masks, from which top ones are selected to build the initial prompt and propagate it across layers.

where $\odot$ denotes element-wise multiplication. To evaluate the predictive utility of each frequency region, we use the reconstructed images $\{\hat{x}_{s,j}\}_{j=1}^{B}$ to compute the task loss for each mask $\mathbf{M}_s$. We then rank all masks $\{\mathbf{M}_s\}_{s=1}^{S}$ in ascending order of their corresponding $\mathcal{L}_{\text{task},s}$. A lower loss implies that the masked frequency region retains more class-discriminative information for the task.

**Phase II — Prompt Initialization.** We select the top-$T$ masks, *i.e.*, the $T$ frequency masks with the lowest task losses $\mathcal{L}_{\text{task},s}$, to initialize the first-layer visual prompt $\mathbf{P}_1^{\text{init}} \in \mathbb{R}^{T \times d}$. Each selected mask $\mathbf{M}_t$ serves as a frequency shortcut, capturing a localized spectral region that preserves strong task-discriminative information. To encode these frequency shortcuts into prompt tokens, we proceed as follows. For each selected mask $\mathbf{M}_t \in \{\mathbf{M}_1, \ldots, \mathbf{M}_T\}$ and each image $x_j \in \{x_1, \ldots, x_B\}$ in a mini-batch, we apply $\mathbf{M}_t$ to the image's Fourier spectrum and reconstruct the filtered image $\hat{x}_{t,j}$ using Equation 2. Each $\hat{x}_{t,j}$ is then passed through the frozen patch embedding module to yield patch tokens $\{\mathbf{p}_{t,j,n} \in \mathbb{R}^{1 \times d}\}_{n=1}^{N}$, where $N$ is the number of image patches. To aggregate the discriminative content emphasized by each frequency shortcut, we employ a token energy-weighted pooling strategy. The intuition is that tokens with higher activation norms tend to carry stronger semantic signals. Specifically, we compute each token's energy and normalize it across the batch:

$$e_{t,j,n} = \left\| \mathbf{p}_{t,j,n} \right\|^2, \quad w_{t,j,n} = \frac{e_{t,j,n}}{\sum_{j'=1}^{B} \sum_{n'=1}^{N} e_{t,j',n'}}. \tag{3}$$

We compute the representative token $\rho_t$ and stack them to obtain the initial prompt matrix $\mathbf{P}_1^{\text{init}}$:

$$\rho_t = \sum_{j=1}^{B} \sum_{n=1}^{N} w_{t,j,n} \cdot \mathbf{p}_{t,j,n}, \quad \mathbf{P}_1^{\text{init}} = \begin{bmatrix} \rho_1^\top \\ \vdots \\ \rho_T^\top \end{bmatrix} \in \mathbb{R}^{T \times d}. \tag{4}$$

To initialize deeper layers, we propagate $\mathbf{P}_1^{\text{init}}$ through the frozen transformer encoder blocks $\{E_i\}_{i=1}^{L-1}$. Starting from the first-layer, each block processes only the prompt and outputs the prompt for the next layer:

$$\mathbf{P}_{i+1}^{\text{init}} = E_i(\mathbf{P}_i^{\text{init}}), \quad \text{for } i = 1, \ldots, L-1. \tag{5}$$

This propagation transfers frequency-aligned semantics through the transformer hierarchy, forming a coherent, task-aware prompt initialization trajectory across all layers.

## 2.3 Optimization via Koopman-Lyapunov discrete dynamical system (KLD)

Following the `MPA` initialization, we use `KLD` to establish cross-layer association of prompts $\{\mathbf{P}_i\}_{i=1}^{L}$. `KLD` is divided into two parts. Firstly, the cross-layer association is established through the Koopman operator, which enables linear evolution in a lifted latent space, facilitating prediction and control from nonlinear data (Mezić, 2013). Then, a Lyapunov-style regularizer is employed to limit the error of the Koopman operator calculation across layers.

**Prompt Linear Projection.** To establish cross-layer associations, KLD first linearly projects the $T$ prompt tokens at each layer into a shared latent space. We introduce a global learnable projection matrix $\mathbf{U} \in \mathbb{R}^{d \times K}$, where $K$ is the dimensionality of the Koopman space. This matrix is initialized using Kaiming-uniform initialization (He et al., 2015), and projects the prompt matrix $\mathbf{P}_i \in \mathbb{R}^{T \times d}$ from layer $i$ into a linear representation $\mathbf{z}_i \in \mathbb{R}^{T \times K}$ as follows:

$$\mathbf{z}_i = \mathbf{P}_i \mathbf{U}. \tag{6}$$

Within this shared latent space, we model prompt evolution using a global shared Koopman operator $\mathbf{K} \in \mathbb{R}^{K \times K}$, which is initialized as the identity matrix.

**Koopman Operator Evolution.** Unlike conventional VPT variants that tune prompts independently per layer, we introduce cross-layer coupling via $\mathbf{K}$. Specifically, we evolve the current state $\mathbf{z}_i$ into the next state $\hat{\mathbf{z}}_{i+1}$ using:

$$\hat{\mathbf{z}}_{i+1} = \mathbf{z}_i \mathbf{K}. \tag{7}$$

This formulation replaces per-layer independence with a global consistent evolution process, as visualized in Figure 3.

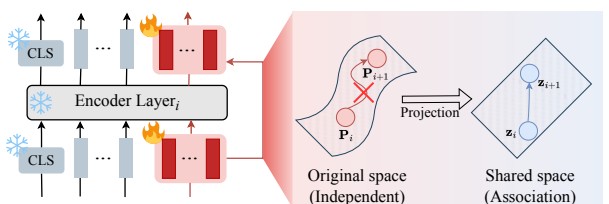

Figure 3: Illustration of Koopman evolution. In standard VPT, prompts at different layers are independent. In contrast, KLD maps each layer's prompt into a shared latent space, where a global shared Koopman operator $\mathbf{K}$ governs their evolution, enabling smooth cross-layer transitions.

**Learning the Koopman Operator.** Effective evolution requires learning both $\mathbf{U}$ and $\mathbf{K}$. The matrix $\mathbf{U}$ learns to map prompts into a space where transitions between layers are approximately linear, while $\mathbf{K}$ captures the shared layer-to-layer dynamics. To jointly learn these parameters, we introduce the *Koopman consistency loss* $\mathcal{L}_{\mathrm{kp}}$, which minimizes the difference between the predicted evolution $\hat{\mathbf{z}}_{i+1}$ and the actual projected state $\mathbf{z}_{i+1}$:

$$\mathcal{L}_{\mathrm{kp}} = \sum_{i=1}^{L-1} \|\mathbf{z}_{i+1} - \hat{\mathbf{z}}_{i+1}\|_F^2 = \sum_{i=1}^{L-1} \|\mathbf{P}_{i+1}\mathbf{U} - \mathbf{P}_i\mathbf{U}\mathbf{K}\|_F^2, \tag{8}$$

where $\|\cdot\|_F$ denotes the Frobenius norm. This loss provides explicit gradient flow across layers by coupling all prompts $\mathcal{P} = \{\mathbf{P}_i\}_{i=1}^{L}$ via the shared parameters $\mathbf{U}$ and $\mathbf{K}$. The gradient with respect to each prompt $\mathbf{P}_i$ is:

$$\frac{\partial \mathcal{L}_{\mathrm{kp}}}{\partial \mathbf{P}_i} = \underbrace{2(\mathbf{P}_i\mathbf{U} - \mathbf{P}_{i-1}\mathbf{U}\mathbf{K})\mathbf{U}^\top}_{\text{preceding-layer consistency: } (i-1)\to i} + \underbrace{2(\mathbf{P}_i\mathbf{U}\mathbf{K} - \mathbf{P}_{i+1}\mathbf{U})(\mathbf{U}\mathbf{K})^\top}_{\text{succeeding-layer consistency: } i\to(i+1)}. \tag{9}$$

This formulation reveals how each layer is pulled toward consistency with both its preceding and succeeding layers, enforcing global smooth and coherent prompt evolution[2].

**A Lyapunov-style Regularizer for Evolution Stability.** Since the linear approximation is imperfect and errors can accumulate across layers (Philipp et al., 2024), we introduce a Lyapunov-style regularizer $\mathcal{L}_{\mathrm{stab}}$ to mitigate error growth and improve stability. Specifically, $\mathcal{L}_{\mathrm{stab}}$ employs a Lyapunov function $V(\mathbf{z}) = \mathrm{tr}(\mathbf{z}\mathbf{Q}\mathbf{z}^\top)$, where $\mathbf{Q} \in \mathbb{R}^{K \times K}$ is a learnable symmetric positive definite matrix. We regard the evolution as stable when the cross-layer error is non-increasing, *i.e.*, when associations between successive layers do not deteriorate (Haber & Ruthotto, 2017). In other words, stability requires $V(\mathbf{z}_{i+1}) \leq V(\mathbf{z_i})$. This condition can be implemented as follows:

$$\mathcal{L}_{\mathrm{stab}} = \sum_{i=1}^{L-1} \max(0, V(\mathbf{z}_{i+1}) - V(\mathbf{z}_i)). \tag{10}$$

As a stability regularizer, $\mathcal{L}_{\mathrm{stab}}$ penalizes increases in the evolution error between successive layers. The penalty is adaptive, and is applied only when cross-layer association degrades, so the evolution remains stable without excessive constraints[3].

---

[2]See Appendix A.2 for full derivation.

[3]The gradient derivation is in Appendix A.3.

Table 1: The speedup($\times$) and image classification accuracy (%) for ViT-Base/16 pretrained on supervised ImageNet-21k. Values are raw accuracy with PAE gains in parentheses.

| Method with PAE | Speedup($\times$) | FGVC | VTAB-1k | | | |
| --- | --- | --- | --- | --- | --- | --- |
| | | | *Natural* | *Specialized* | *Structured* | Mean Total |
| Full Fine-tune | - | 88.54 | 75.88 | 83.36 | 47.64 | 68.96 |
| VPT (Jia et al., 2022) | 1.78 | 89.11 +1.91 | 78.48 +3.25 | 82.43 +2.09 | 54.98 +3.30 | 71.96 +2.88 |
| E2VPT (Han et al., 2023) | 1.65 | 89.22 +1.74 | 80.01 +1.38 | 84.43 +1.33 | 57.39 +2.34 | 73.94 +1.68 |
| LPT (Dong et al., 2023) | 1.44 | 89.94 +1.38 | 79.24 +1.77 | 83.40 +1.62 | 57.39 +2.02 | 73.34 +1.81 |
| VQT (Tu et al., 2023) | 1.52 | 89.41 +1.21 | 79.46 +2.93 | 82.23 +3.11 | 57.47 +2.36 | 73.05 +2.80 |
| VFPT (Zeng et al., 2024) | 1.27 | 89.24 +2.24 | 81.35 +0.72 | 84.93 +1.03 | 60.19 +0.77 | 75.39 +0.94 |
| SA2VP (Pei et al., 2024) | 1.60 | 90.08 +1.12 | 80.97 +1.89 | 85.73 +0.85 | 60.80 +2.25 | 75.83 +1.66 |
| ProVP (Xu et al., 2025) | 1.19 | 89.56 +1.62 | 80.35 +1.98 | 84.07 +1.34 | 60.31 +1.41 | 74.91 +1.57 |
| BPT (Wang et al., 2025b) | 1.37 | 90.86 +1.35 | 80.24 +2.22 | 84.45 +1.88 | 60.39 +1.66 | 75.02 +1.92 |

**Total Training Objective.** Finally, our complete model is trained end-to-end by minimizing a total objective function that combines the primary downstream task loss, $\mathcal{L}_{\text{task}}$, with our two dynamics-based regularizers:

$$\mathcal{L}_{\text{total}} = \mathcal{L}_{\text{task}} + \alpha\mathcal{L}_{\text{kp}} + \beta\mathcal{L}_{\text{stab}}, \tag{11}$$

where $\alpha$ and $\beta$ are hyperparameters that balance the enforcement of coherent dynamics and stability against the task-aware objective.

## 3 EXPERIMENTS

### 3.1 EXPERIMENTAL SETUP

**Datasets** We conduct experiments on 2 classification benchmarks: Fine-Grained Visual Classification (FGVC) and VTAB-1k (Zhai et al., 2019). VTAB-1k comprises 19 tasks across *Natural*, *Specialized*, and *Structured* domains, with a constrained budget of 1,000 labeled training samples per task. The FGVC benchmark refers to a collection of five fine-grained classification datasets. Additionally, we evaluate semantic segmentation performance on the ADE20K (Zhou et al., 2017). All 25 datasets follow the training and testing split protocols defined in VPT (Jia et al., 2022)[4].

**Baseline Models** To evaluate the effectiveness of PAE, we incorporate PAE into several state-of-the-art VPT variants, including structured prompt designs (VPT (Jia et al., 2022), E2VPT (Han et al., 2023), VQT (Tu et al., 2023), ProVP (Xu et al., 2025), BPT (Wang et al., 2025b)), adaptive prompting (LPT (Dong et al., 2023)), and perception-driven priors (VFPT (Zeng et al., 2024), SA2VP (Pei et al., 2024)). For each method, we report both its

Table 2: Results of ADE20K datasets with ViT-L. SS/MS denote single/multi-scale inference. Values are mIoU with PAE gains.

| Methods | Speedup($\times$) | mIoU-SS | mIoU-MS |
| --- | --- | --- | --- |
| Full-tuning | - | 47.60 | 49.18 |
| SPT-LoRA | - | 45.40 | 47.50 |
| VPT | 1.29$\times$ | 44.08 + 2.73 | 46.01 + 1.96 |
| E2VPT | 1.18$\times$ | 44.61 + 2.32 | 46.56 + 2.84 |
| VFPT | 1.15$\times$ | 45.32 + 2.75 | 47.17 + 2.09 |

original performance and the enhanced version with PAE, denoted as "Baseline +$\Delta$", to isolate the contribution of our framework. Then, we quantify convergence rate with speedup: the baseline epochs to reach its best val acc divided by the epochs required after using PAE. Values >1 mean faster convergence. In addition, we include *Full Fine-tuning*, where all parameters are updated. The experiments are conducted by 2 Transformer architectures: 3 variants (ViT-B/16, -L/16, and -H/14,) of ViT (Dosovitskiy et al., 2020) and Swin-B (Liu et al., 2021), where are pre-trained on ImageNet-21k (Deng et al., 2009). Moreover, we evaluate a self-supervised ViT B/16 backbone: MAE (He et al., 2022), and a segmentation ViT L/16 backbone: SETR (Zheng et al., 2021).

**Implementation Details** In MPA, we set the window size $w$=16 with stride $r$=8. We randomly selected a input mini-batch to discover frequency shortcuts. Then, the dimension of Koopman operator was set to 256. The weights were set to $\alpha = 0.5$ for $\mathcal{L}_{\text{kp}}$ and $\beta = 0.2$ for $\mathcal{L}_{\text{stab}}$[5]. Cross-entropy (CE) loss was used for $\mathcal{L}_{\text{task}}$. After adding PAE, the initial learning rate was set to 0.25, which was different from the larger learning rate of VPT variants. The batch size was set to 128. All experiments were conducted on an NVIDIA A800 GPU.

---

[4]Detailed descriptions and statistics of these datasets are provided in Appendix A.4.

[5]More ablation studies on window size, stride, loss weights are provided in Appendix A.5.

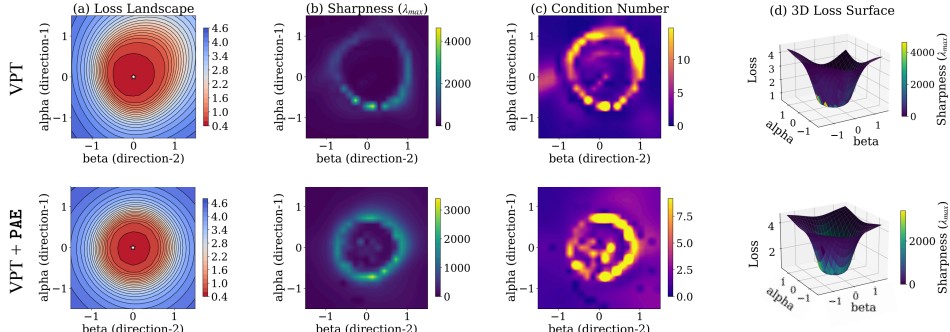

Figure 4: Loss landscape comparisons (Li et al., 2018) show (left to right): 2-D loss contours, sharpness (max Hessian eigenvalue), anisotropy (Hessian condition number), and a 3-D surface colored by curvature.

## 3.2 RESULTS

The results[6] on the FGVC and VTAB-1k benchmarks are summarized in Table 1. With only a slight increase in preprocessing time, MPA completes initialization in just 74.17 seconds, roughly equivalent to 5.3 training epochs. Incorporating PAE consistently boosts performance across all settings, underscoring its versatility as a prompt-agnostic enhancement with no inference-time overhead. For instance, integrating PAE into VPT (Jia et al., 2022) raises FGVC accuracy on 91.02% (89.11%+**1.91%**) and VTAB-1k mean accuracy on 74.84% (71.96% +**2.88%**) with a convergence speedup of 1.78×. Similarly, when applied to SA2VP (Pei et al., 2024), it improves VTAB-1k performance on 77.49% (75.83%+**1.66%**), and achieves a 1.60× speedup in terms of convergence. Furthermore, adding PAE brings improvements on several VPT variants on ADE20K (Table 2). Specifically, adding PAE to VPT, E2VPT, and VFPT increases mIoU by about 2-3% under both single- and multi-scale evaluation, while also speeding up by roughly 1.15 to 1.29 ×. These results show that PAE is still beneficial for dense prediction tasks such as semantic segmentation.

Furthermore, we visualize the loss landscape on a 2D subspace and report curvature statistics in Figure 4. Applying PAE results in a substantially larger low-loss region with near-circular contours, indicating reduced anisotropy. The Hessian-based sharpness map shows diminished high-curvature rings when PAE is applied, and the condition number heatmap displays uniformly lower values, both suggesting a more isotropic loss surface. The 3D surface plot further illustrates that VPT with PAE converges to a wider and flatter minimum, whereas the baseline VPT remains sharp and narrow. Then, we employ Grad-CAM (Selvaraju et al., 2017) to visualize the impact of integrating PAE

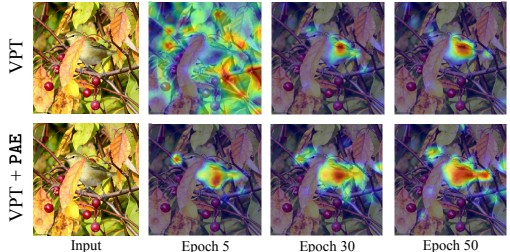

Figure 5: Grad-CAM visualizations at early training stages (epochs 5, 30, and 50) show that VPT+PAE rapidly concentrates attention on class-discriminative regions and stabilizes the saliency patterns, showing faster convergence than VPT.

into VPT on the CUB dataset, which consists of 200 fine-grained bird species. As shown in Figure 5, at epoch 5, VPT displays diffuse and non-discriminative attention, while VPT+PAE already focuses on task-relevant regions such as the *wing* and *beak* when classifying the Tennessee warbler. By epoch 30, VPT begins to localize the bird, but VPT+PAE delineates its full body with greater precision. At epoch 50, VPT+PAE produces sharper and more consistent saliency maps, indicating improved discriminative localization. These observations collectively indicate that using PAE leads to flatter, more stable minima, enhances discriminative localization, and accelerates convergence.

To assess the efficiency and scalability of PAE across diverse architectures and model sizes, we evaluate VPT, E2VPT, VFPT, and VPT+PAE across three VTAB-1k groups on 4 backbones that vary in different *scale* and *architecture* (ViT-B/16, Swin-B, ViT-L/16, ViT-H/14) in Figure 6. Across all three groups, VPT+PAE consistently improves over VPT on every backbone and remains competitive

---

[6]Complete per-dataset scores are provided in Appendix A.7.

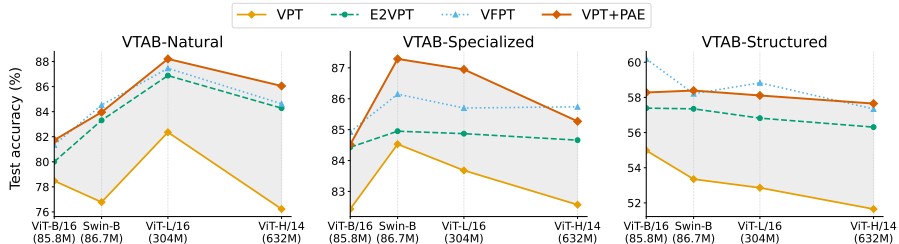

Figure 6: VPT, E2VPT, VFPT, and VPT+PAE across three VTAB groups and four backbones (ViT-B/16 → ViT-H/14). The shaded area marks the gap between VPT and VPT+PAE.

with or better than E2VPT and VFPT, indicating that PAE scales smoothly from base to huge ViTs and transfers to other architectures, such as Swin-B.

Beyond performance, we further analyze cross-layer prompt interactions by visualizing prompt CKA (Kornblith et al., 2019) across layers on an MAE backbone, following Yoo et al. (2023). As shown in Figure 7, VPT and VFPT exhibit large blocks of high CKA, reflecting highly redundant prompts and weak depth dependent structure. GatePT (Yoo et al., 2023) slightly smooths these correlations but still maintains globally high similarity across most layers. In contrast, VPT+PAE produces a sharp diagonal band in the CKA matrix, where similarity is strongest locally and gradually decreases with layer distance. This pattern indicates a progressive, depth-aware evolution of the prompt state rather than a globally entangled representation. The diagonal structure aligns with our formulation: a single shared Koopman operator governs prompt dynamics and induces a stable, forward-evolving trajectory across layers.

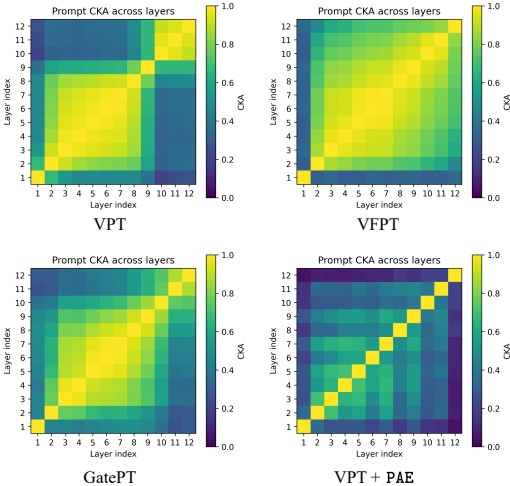

Figure 7: Prompt CKA (Kornblith et al., 2019) across layers for different VPT variants on MAE.

We additionally analyze how PAE behaves on classes with different levels of intra-class variance (Cong et al., 2024) on CUB-200-2011. For each class, we compute its intra-class variance and sort classes along the x-axis from low to high variance. In Fig. 8(a), we plot the per-class accuracy of VPT+PAE against intra-class variance. We observe a mild negative Pearson correlation (corr = -0.290), this shows high-variance classes tend to have lower accuracy, confirming that intra-class variance is a good proxy for class difficulty. In Fig. 8(b), we plot the relative accuracy gain of

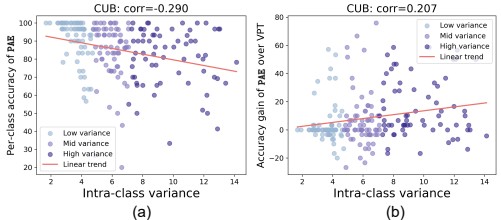

Figure 8: (a) VPT+PAE per-class accuracy vs. intra-class variance. (b) PAE improvement over VPT vs. intra-class variance.

VPT+PAE over vanilla VPT for each class, using the same ordering by intra-class variance. Here we observe a mild positive correlation (corr = 0.207), this shows although high-variance classes are absolutely harder, they receive larger relative improvements from PAE. This indicates that PAE provides the greatest benefit on the difficult, high-variance categories where VPT struggles most.

### 3.3 ABLATION STUDIES

We conduct an ablation study to dissect the PAE framework and quantify the contributions of core components. Results are presented in Table 3. The study reveals two key findings. First, the MPA strategy is the most critical contributor to performance. When used in isolation, MPA yields a +2.06% improvement on the VTAB-1k mean score over the baseline. This gain significantly exceeds that of using only the Koopman consistency loss ($\mathcal{L}_{kp}$). Second, $\mathcal{L}_{kp}$ and the Lyapunov stability loss $\mathcal{L}_{stab}$

Table 3: Ablation of `PAE` core elements on ViT-Base/16 using VPT (Jia et al., 2022) as baseline.

| MPA(init) | KLD ($\mathcal{L}_{\text{kp}}$) | KLD ($\mathcal{L}_{\text{stab}}$) | FGVC | VTAB-1k | | | |
|---|---|---|---|---|---|---|---|
| | | | | *Natural* | *Specialized* | *Structured* | Mean |
| ✗ | ✗ | ✗ | 89.11 | 78.48 | 82.43 | 54.98 | 71.96 |
| ✓ | ✗ | ✗ | 89.63 | 81.35 | 84.07 | 56.64 | 74.02 |
| ✗ | ✓ | ✗ | 90.56 | 80.27 | 83.28 | 55.82 | 73.13 |
| ✗ | ✓ | ✓ | 90.78 | 81.46 | 83.85 | 57.96 | 74.42 |
| ✓ | ✓ | ✓ | **91.02** | **81.73** | **84.52** | **58.28** | **74.84** |

Table 4: Ablation studies on ViT-B/16, using VPT with `PAE`: (Left) Comparison across different prompt initialization strategies. (Right) Variants of frequency shortcut injection.

| (a) Prompt Initialization Strategies | | | | | | (b) Frequency–Shortcut Injection Variants | | | | | |
|---|---|---|---|---|---|---|---|---|---|---|---|
| Method | FGVC | VTAB-1k | | | | Variant | FGVC | VTAB-1k | | | |
| | | *Natural* | *Specialized* | *Structured* | Mean | | | *Natural* | *Specialized* | *Structured* | Mean |
| Random | 89.11 | 78.48 | 82.43 | 54.98 | 71.96 | Shallow | 89.37 | 79.14 | 82.36 | 55.47 | 72.32 |
| Xavier | 89.55 | 80.67 | 82.32 | 53.91 | 72.30 | Copy | 89.83 | 80.05 | 83.11 | 56.34 | 73.17 |
| UnPro | 90.23 | 80.46 | 83.18 | 57.89 | 73.85 | Layer-wise | 89.86 | 81.27 | 83.68 | 57.93 | 74.29 |
| MPA (Ours) | **91.02** | **81.73** | **84.52** | **58.28** | **74.84** | Shared (Ours) | **91.02** | **81.73** | **84.52** | **58.28** | **74.84** |

are also essential and demonstrate strong synergy. While $\mathcal{L}_{\text{kp}}$ alone yields notable improvement, adding $\mathcal{L}_{\text{stab}}$ results in a further +1.29% gain. The full `PAE` framework, integrating all three components, achieves the highest performance, confirming that each part plays a complementary role in addressing the underlying optimization challenges.

To further validate the effectiveness of `MPA`, we compare it with several commonly used initialization strategies, as shown in Table 4(a). For fair comparison, all methods use the $\mathcal{L}_{\text{kp}}$ and $\mathcal{L}_{\text{stab}}$. Random initialization yields a VTAB-1k mean score of 71.96%, while Xavier initialization (Xu et al., 2025) slightly improves this to 72.30% but lacks task-aware alignment. A stronger baseline uses features derived from

Table 5: Ablation on different input mini-batches using VPT+PAE.

| Mini-batch | FGVC | VTAB-1k | | | | p-value |
|---|---|---|---|---|---|---|
| | | *Natural* | *Specialized* | *Structured* | Mean | |
| $B^{(1)}$ | 91.02 | 81.73 | 84.52 | 58.28 | 74.84 | - |
| $B^{(2)}$ | 90.62 | 81.47 | 84.54 | 57.95 | 74.65 | 0.081 |
| $B^{(3)}$ | 91.07 | 81.39 | 84.23 | 57.92 | 74.51 | 0.092 |
| $B^{(4)}$ | 90.98 | 82.01 | 84.96 | 58.66 | 75.21 | 0.089 |
| $B^{(5)}$ | 90.64 | 81.34 | 84.14 | 57.91 | 74.46 | 0.008 |

Unsupervised Prototypes (Wang et al., 2024), achieving 73.85% by leveraging general-purpose visual representations. In contrast, `MPA` consistently outperforms all these alternatives, reaching 74.84%. This highlights its unique advantage: rather than relying on generic representations, `MPA` conducts a task-aware search to identify frequency shortcuts that are most relevant to the target task.

We further investigate how prompts are propagated across layers in `MPA` in Table 4(b). After generating frequency shortcut masks, `MPA` constructs the first-layer prompt and propagates it through the frozen backbone to initialize prompts in subsequent layers—achieving the best performance at 74.84% VTAB-1k mean. This outperforms two alternative designs. First, copying the first-layer prompt to all layers yields 73.17%, indicating that building a layer-specific prompt hierarchy is crucial. Second, performing a layer-wise independent search achieves 74.29%, which still underperforms `MPA`. These findings show the effectiveness of our single-search and propagate strategy.

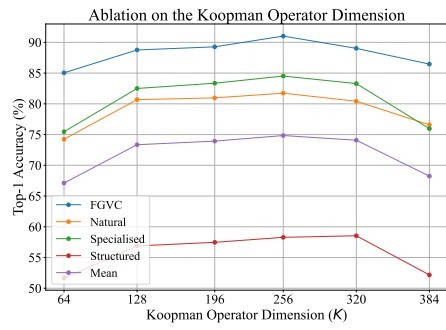

Figure 9: Ablation on the Koopman operator dimension in VPT+PAE on ViT-B/16.

With batch size $B$ and hyperparameters fixed ($w$=16, $r$=8, $B$=128), we examine the effect of randomly selected input batches on `MPA`, as shown in Table 5. Batch $B^{(1)}$ is used as the baseline, and paired $t$-tests are performed against it. Deviations across task categories remain small: FGVC $\leq$ 0.40, *Natural* $\leq$ 0.39, *Specialized* $\leq$ 0.44, and *Structured* $\leq$ 0.38, yielding $p \in [0.008, 0.092]$. These results demonstrate that the randomness introduced by selecting different input batches for `MPA` has a negligible impact on performance.

Finally, we study the effect of the Koopman operator's latent dimension $K$. As shown in Figure 9. $K$ controls the expressivity of the linear dynamics model. The results show a clear trade-off: a small dimension (*e.g.*, $K = 64$) leads to underfitting and poor performance (67.11%). As $K$ increases, performance improves, peaking at $K = 256$ with a mean VTAB-1k score of 74.84%. However, further increasing $K$ to 384 degrades performance (68.24%), suggesting that an overly large latent

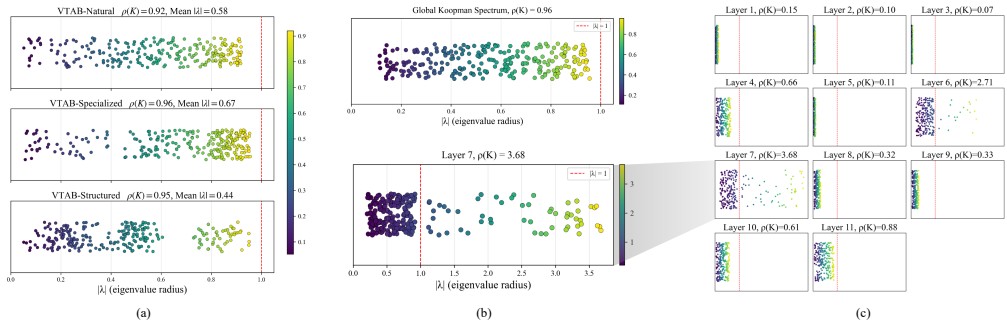

Figure 10: (a) Global Koopman spectra computed for the Natural, Specialized, and Structured subsets of VTAB-1k. (b) Comparison between the spectrum of the global Koopman operator (top) and that of the layer-wise operator at layer 7 (bottom). (c) The layer-wise design exhibits the individual spectra for each layer-specific operator.

space introduces optimization challenges and increases the risk of overfitting. Based on this analysis, we set $K = 256$ as the default configuration.

## 4  ANALYSIS

According to classical linear systems theory, Eq.7 is globally asymptotically stable if the spectral radius $\rho(\mathbf{K}) < 1$ (Guglielmi & Overton, 2011). In Koopman theory, eigenvalue magnitude dictates persistence: values near one decay slowly, while small ones vanish quickly (Budišić et al., 2012; Williams et al., 2015). In practice, we analyze the spectrum via $\mathbf{K}v = \lambda v$. Specifically, Eq.8 ($\mathcal{L}_{\text{kp}}$) is used in the shared latent space, and Eq.10 ($\mathcal{L}_{\text{stab}}$) penalizes growth of the energy. These losses favor contraction without rotation and concentrate the spectrum on the positive real axis.

We first compare the global Koopman spectra learned by PAE on VTAB-1k benchmark from the Natural, Specialized, and Structured groups. As shown in Fig. 10(a), all three learned global operators are stable with $\rho(\mathbf{K}) < 1$, but their spectra exhibit clear group-specific patterns. Specifically, Natural tasks show a moderate mean radius (mean$|\lambda| = 0.58$), indicating that PAE only needs to gently reshape pretrained features when transferring to other natural-image tasks. Specialized tasks attain the largest mean radius (mean$|\lambda| = 0.67$) with slow modes near unity, suggesting long effective memory for strong domain shifts. This helps cope with stronger domain shifts such as histopathology and remote sensing. In contrast, structured tasks yield the smallest radius (mean$|\lambda| = 0.44$), consistent with aggressive damping of texture variations. These group-dependent spectra reveal dynamical patterns in how PAE adapts its cue dynamics to different downstream tasks.

Next, we compare the spectrum of the global operator learned by PAE against a layer-wise design, in which an independent learnable operator $\mathbf{K}_i$ is assigned to each layer transition $i \rightarrow i+1$. Fig. 10(b) shows that the global operator concentrates eigenvalues along the positive real axis with $\rho(\mathbf{K}) < 1$, ensuring coherent evolution. In contrast, the eigenvalue radius at layer 7 exceeds 3, reaching 3.68. As shown in Fig. 10(c), the layer-wise variant yields highly heterogeneous dynamics. Several layers exhibit large spectral radii (eigenvalue radius $\geq 1$), introducing mismatched time scales and unstable modes that are significantly harder to regularize.

## 5  CONCLUSION

In this paper, we reframe visual prompt tuning as a discrete dynamical system and introduce PAE, coupling MPA and KLD. MPA employs a supervision mechanism to find task-aware frequency shortcuts, which convert into initial prompts. KLD applies a shared Koopman evolution with a Lyapunov-style regularizer. This establishes cross-layer dependencies explicit, stabilizes gradients, accelerates convergence, and improves performance. Across multiple variants and image-classification benchmarks, PAE improves accuracy and convergence. In particular, PAE remains prompt-agnostic: it drops into various VPT variants with no backbone changes and no inference-time overhead, offering a practical and scalable path to improve visual prompt tuning. **Limitation:** MPA may be less suitable for low- or zero-label regimes unless appropriate surrogate signals are available.

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

# A  APPENDIX

## A.1  THE USES OF LLM

We used a large language model (LLM) only for language polishing (grammar, clarity, and style). All ideas, analyses, and conclusions are the authors' own, and the authors conducted the final review and approval.

## A.2  GRADIENT FOR THE KOOPMAN CONSISTENCY LOSS

Here we provide the detailed derivation for the gradient of the Koopman consistency loss $\mathcal{L}_{\mathrm{kp}}$ with respect to an intermediate prompt $\mathbf{P}_i$. The loss is given by:

$$\mathcal{L}_{\mathrm{kp}} = \sum_{i=1}^{L-1} \big\| \underbrace{\mathbf{z}_{i+1}}_{\mathbf{P}_{i+1}\mathbf{U}} - \underbrace{\mathbf{z}_i\mathbf{K}}_{\hat{\mathbf{z}}_{i+1}} \big\|_F^2. \tag{12}$$

The prompt $\mathbf{P}_i$ (and its Koopman representation $\mathbf{z}_i = \mathbf{P}_i\mathbf{U}$) influences two adjacent terms in this sum: the term for the transition from $i - 1$ to $i$, which we denote $l_{i-1} = \|\mathbf{z}_i - \mathbf{z}_{i-1}\mathbf{K}\|_F^2$, and the term for the transition from $i$ to $i + 1$, denoted $l_i = \|\mathbf{z}_{i+1} - \mathbf{z}_i\mathbf{K}\|_F^2$.

The gradient is the sum of the gradients of these two terms. First, we compute the gradient with respect to $\mathbf{z}_i$:

$$\frac{\partial l_{i-1}}{\partial \mathbf{z}_i} = 2(\mathbf{z}_i - \mathbf{z}_{i-1}\mathbf{K}) \tag{13}$$

$$\frac{\partial l_i}{\partial \mathbf{z}_i} = -2(\mathbf{z}_{i+1} - \mathbf{z}_i\mathbf{K})\mathbf{K}^\top \tag{14}$$

Combining these gives the full gradient with respect to $\mathbf{z}_i$:

$$\frac{\partial \mathcal{L}_{\mathrm{kp}}}{\partial \mathbf{z}_i} = 2(\mathbf{z}_i - \mathbf{z}_{i-1}\mathbf{K}) - 2(\mathbf{z}_{i+1} - \mathbf{z}_i\mathbf{K})\mathbf{K}^\top = 2(\mathbf{z}_i - \hat{\mathbf{z}}_i) - 2(\mathbf{z}_{i+1} - \hat{\mathbf{z}}_{i+1})\mathbf{K}^\top. \tag{15}$$

Finally, using the chain rule with $\mathbf{z}_i = \mathbf{P}_i\mathbf{U}$, we obtain the gradient with respect to the prompt $\mathbf{P}_i$:

$$\frac{\partial \mathcal{L}_{\mathrm{kp}}}{\partial \mathbf{P}_i} = \frac{\partial \mathcal{L}_{\mathrm{kp}}}{\partial \mathbf{z}_i}\frac{\partial \mathbf{z}_i}{\partial \mathbf{P}_i} = \big[2(\mathbf{z}_i - \hat{\mathbf{z}}_i) - 2(\mathbf{z}_{i+1} - \hat{\mathbf{z}}_{i+1})\mathbf{K}^\top\big]\mathbf{U}^\top. \tag{16}$$

This final gradient provides the direct supervisory signal used to update the prompts.

## A.3  GRADIENT FOR THE LYAPUNOV STABILITY REGULARIZER

The Lyapunov stability regularizer is defined as:

$$\mathcal{L}_{\mathrm{stab}} = \sum_{i=1}^{L} \max(0, \Delta V_i), \quad \text{where} \quad \Delta V_i \triangleq V(\mathbf{z}_{i+1}) - V(\mathbf{z}_i). \tag{17}$$

The Lyapunov function is $V(\mathbf{z}) = \mathrm{tr}(\mathbf{z}\mathbf{Q}\mathbf{z}^\top)$. Its gradient with respect to $\mathbf{z}$ is $\partial V(\mathbf{z})/\partial \mathbf{z} = 2\mathbf{z}\mathbf{Q}$, given that $\mathbf{Q}$ is symmetric.

To handle the non-differentiability of the $\max(0, x)$ function at $x = 0$, we can use an indicator function in the derivation. Let $\eta_i = \mathbf{1}_{\{\Delta V_i > 0\}}$ be an indicator that is 1 if the error increases from step $i$ to $i + 1$, and 0 otherwise. The loss can be expressed as $\mathcal{L}_{\mathrm{stab}} = \sum_{i=1}^{L} \eta_i \Delta V_i$.

Let's compute the gradient with respect to an intermediate state $\mathbf{z}_i$ (where $0 < i < L$). The state $\mathbf{z}_i$ influences two terms in the sum: $\eta_{i-1}\Delta V_{i-1} = \eta_{i-1}(V(\mathbf{z}_i) - V(\mathbf{z}_{i-1}))$ and $\eta_i\Delta V_i = \eta_i(V(\mathbf{z}_{i+1}) - V(\mathbf{z}_i))$. The gradient is therefore:

$$\begin{aligned}
\frac{\partial \mathcal{L}_{\mathrm{stab}}}{\partial \mathbf{z}_i} &= \frac{\partial}{\partial \mathbf{z}_i}\Big[\eta_{i-1}\big(V(\mathbf{z}_i) - V(\mathbf{z}_{i-1})\big)\Big] + \frac{\partial}{\partial \mathbf{z}_i}\Big[\eta_i\big(V(\mathbf{z}_{i+1}) - V(\mathbf{z}_i)\big)\Big] \\
&= \eta_{i-1}\frac{\partial V(\mathbf{z}_i)}{\partial \mathbf{z}_i} - \eta_i\frac{\partial V(\mathbf{z}_i)}{\partial \mathbf{z}_i} \\
&= 2\eta_{i-1}\mathbf{z}_i\mathbf{Q} - 2\eta_i\mathbf{z}_i\mathbf{Q} \\
&= 2(\eta_{i-1} - \eta_i)\mathbf{z}_i\mathbf{Q}.
\end{aligned} \tag{18}$$

This resulting gradient for $\mathbf{z}_i$ is non-zero only if the stability condition changes at the boundary of step $i$ (i.e., violated for step $i-1$ but not for step $i$, or vice-versa).

Finally, using the chain rule with $\mathbf{z}_i = \mathbf{P}_i\mathbf{U}$, we obtain the gradient with respect to the prompt $\mathbf{P}_i$:

$$\frac{\partial \mathcal{L}_{\text{stab}}}{\partial \mathbf{P}_i} = \frac{\partial \mathcal{L}_{\text{stab}}}{\partial \mathbf{z}_i}\frac{\partial \mathbf{z}_i}{\partial \mathbf{P}_i} = 2(\eta_{i-1} - \eta_i)\mathbf{z}_i\mathbf{Q}\mathbf{U}^\top. \tag{19}$$

This provides the corrective signal to update the prompts, active only when necessary to enforce the error constraint.

## A.4 DETAILED DATASET DESCRIPTIONS

We provide comprehensive statistics for the datasets used in our evaluation, corresponding to the experimental setup described in the main text.

**Fine-Grained Visual Classification (FGVC).**   This benchmark includes five datasets that require distinguishing between highly similar subordinate categories. The specific data splits are as follows:

- **CUB-200-2011** (Wah et al., 2011): 200 classes (5,394 train, 600 val, 5,794 test).

- **NABirds** (Van Horn et al., 2015): 55 classes (21,536 train, 2,393 val, 24,633 test).

- **Oxford Flowers-102** (Nilsback & Zisserman, 2008): 102 classes (1,020 train, 1,020 val, 6,149 test).

- **Stanford Dogs** (Khosla et al., 2011): 120 classes (10,800 train, 1,200 val, 8,580 test).

- **Stanford Cars** (Gebru et al., 2017): 196 classes (7,329 train, 815 val, 8,041 test).

**VTAB-1k Benchmark.**   The Visual Task Adaptation Benchmark (VTAB-1k) (Zhai et al., 2019) limits the label budget to 1,000 training samples per task (800 for training, 200 for validation). It consists of 19 tasks grouped into three categories:

- **Natural**: CIFAR-100 (10,000 test), Caltech101 (6,084 test), DTD (1,880 test), Flowers102 (6,149 test), Pets (3,669 test), SVHN (26,032 test), and SUN397 (21,750 test).

- **Specialized**: Patch Camelyon (32,768 test), EuroSAT (5,400 test), RESISC45 (6,300 test), and Retinopathy (42,670 test).

- **Structured**: CLEVR/count (15,000 test), CLEVR/distance (15,000 test), DMLab (22,735 test), KITTI/distance (711 test), dSprites/location (73,728 test), dSprites/orientation (73,728 test), SmallNORB/azimuth (12,150 test), and SmallNORB/elevation (12,150 test).

**Semantic Segmentation.**   For dense prediction tasks, we utilize the **ADE20K** (Zhou et al., 2017) dataset. Train/val splits were followed the same split protocol as in our main experimental setup.

## A.5 ABLATION STUDIES ON HYPERPARAMETERS

As shown in Table 6, the frequency-grid hyperparameters show that $(w{=}16,\ r{=}8)$ attains the highest cross-benchmark Mean $= 74.84$ and is best (or tied) on FGVC 91.02 and *Natural* 81.73. Although per-track optima differ—*Specialized* peaks at $(16, 12)$ with 85.07 and *Structured* peaks at $(24, 12)$ with 59.11, we choose $(w{=}16,\ r{=}8)$ as the final configuration *because it maximizes the Mean*, which is our selection criterion across benchmarks. Unless otherwise stated, subsequent experiments use $\alpha{=}0.5,\ \beta{=}0.2$ and $(w{=}16,\ r{=}8)$.

As shown in Table 7, introducing moderate loss weights consistently improves over the no-auxiliary baseline ($\alpha{=}\beta{=}0$, Mean $= 71.96$). The best average appears at $\alpha{=}0.5,\ \beta{=}0.2$ with Mean $= 74.84$ (+2.88), alongside gains on all tracks: FGVC 91.02 (+1.91), *Natural* 81.73 (+3.25), *Specialized* 84.52 (+2.09), and *Structured* 58.28 (+3.30). Further increasing $\alpha$ (e.g., $\alpha{=}0.9,\ \beta{=}0.3$) degrades the average (73.52), indicating over-regularization.

Table 6: Ablation on frequency window size $w$ and stride $r$ for ViT-B/16 with VPT+PAE. *Bold* denotes the best in column.

| $w$ | $r$ | FGVC | VTAB-1k | | | |
|---|---|---|---|---|---|---|
| | | | *Natural* | *Specialized* | *Structured* | Mean |
| 8 | 4 | 89.28 | 79.76 | 82.71 | 55.29 | 72.59 |
| 16 | 8 | **91.02** | **81.73** | 84.52 | 58.28 | **74.84** |
| 16 | 12 | 90.37 | 79.84 | **85.07** | 57.73 | 74.21 |
| 24 | 12 | 90.41 | 80.43 | 82.96 | **59.11** | 74.16 |
| 24 | 16 | 90.15 | 79.97 | 83.56 | 58.62 | 74.05 |

Table 7: Ablation on loss weights $(\alpha, \beta)$ for ViT-B/16 with VPT+PAE. *Bold* denotes the best in column.

| $\alpha$ | $\beta$ | FGVC | VTAB-1k | | | |
|---|---|---|---|---|---|---|
| | | | *Natural* | *Specialized* | *Structured* | Mean |
| 0.0 | 0.0 | 89.11 | 78.48 | 82.43 | 54.98 | 71.96 |
| 0.1 | 0.1 | 89.58 | 79.02 | 83.01 | 55.67 | 72.57 |
| 0.3 | 0.1 | 90.72 | 80.31 | 84.24 | 57.83 | 74.13 |
| 0.5 | 0.2 | **91.02** | 81.73 | **84.52** | **58.28** | **74.84** |
| 0.7 | 0.2 | 90.63 | **82.18** | 84.17 | 58.09 | 74.81 |
| 0.9 | 0.3 | 89.37 | 79.95 | 82.90 | 57.71 | 73.52 |

## A.6 SENSITIVITY ANALYSIS OF CONVERGENCE SPEED.

To rigorously evaluate the robustness of our hyperparameter selection, we analyze the impact of the regularization weights $(\alpha, \beta)$ on the convergence speed across four distinct benchmarks: FGVC, VTAB-Natural, VTAB-Specialized, and VTAB-Structured. As illustrated in Figure 11, a consistent **U-shaped correlation** is observed between the regularization strength and the number of epochs required for convergence across all task groups.

Specifically, the experimental results reveal three key regimes centering around an optimal configuration. The fastest convergence is uniformly achieved at the **"Sweet Spot"** setting of $(\alpha = 0.5, \beta = 0.2)$. In this region, the stability induced by PAE allows for aggressive optimization, reducing the training duration by approximately $40\% \sim 50\%$ compared to the baseline $(\alpha = 0, \beta = 0)$. Deviating from this optimum compromises efficiency. In the **Under-regularization** regime (e.g., $0.1, 0.1$), weak constraints fail to effectively suppress high-frequency gradient oscillations, leaving the optimization trajectory jagged and resulting in negligible epoch reduction. Conversely, **Over-regularization** (e.g., $0.9, 0.3$) leads to a sharp rebound in training costs due to excessive constraints. This is particularly evident in the *FGVC* group, where "over-smoothing" prompts detrimentally affects the learning of fine-grained discriminative features, prolonging convergence to 96 epochs.

Notably, the *Natural* and *Specialized* groups exhibit the steepest improvements, benefiting most from the stabilized dynamics. While the *Structured* tasks show a flatter improvement curve due to their inherent optimization difficulty, they still achieve optimal efficiency at the same parameter configuration, confirming the universality of our default setup.

## A.7 FULL EXPERIMENTAL RESULTS

For FGVC in Figure 8, we report per-dataset top-1 accuracy and convergence epoch for all evaluated methods, including baseline VPT variants and their PAE-augmented counterparts. For VTAB-1k, we provide per-task scores for every task: Natural in Figure 9, Specialized in Figure 10, and Structured in Figure 11, along with the group means and the overall mean. Training and evaluation settings match those in the main text unless otherwise noted.

We compare both variants of VPT with other commonly used fine-tuning protocols:

(a) Fully fine-tune all backbone and classification head parameters.

(b) Methods that focus on the classification head. They treat the pre-trained backbone as a feature extractor, whose weights are fixed during tuning:

   - Linear: only use a linear layer as the classification head.

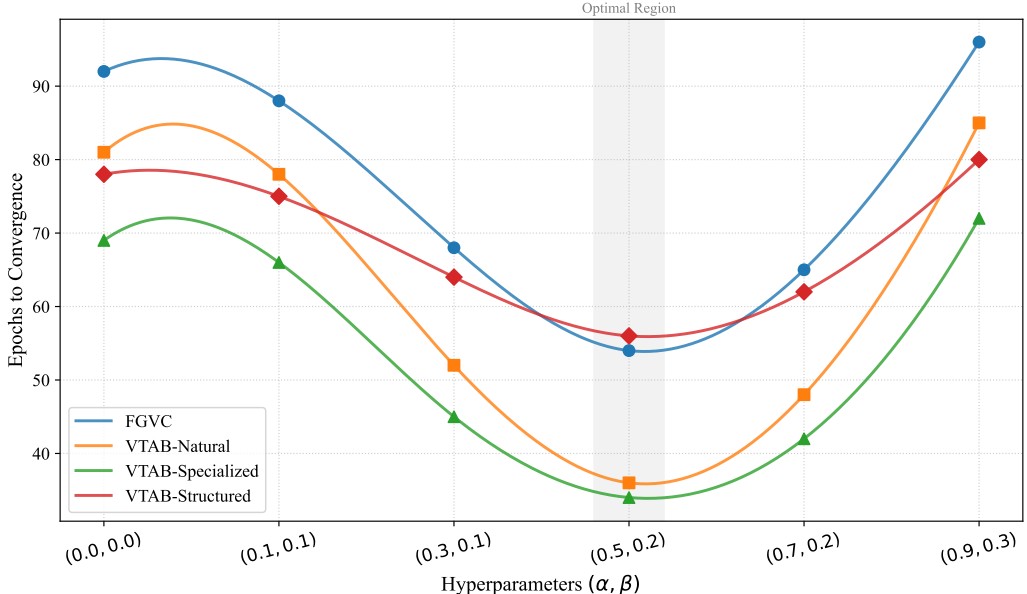

Figure 11: Sensitivity analysis of convergence speed with respect to hyperparameters $(\alpha, \beta)$.

- Partialft-$k$: fine-tune the last $k$ layers of the backbone while freezing the others, as adopted in (Yosinski et al., 2014; Zhang et al., 2016; Noroozi & Favaro, 2016; He et al., 2022). This redefines the boundary between the backbone and the classification head.
- Mlp-$k$: utilize a multilayer perceptron (MLP) with $k$ layers, instead of a linear layer, as the classification head.

(c) Methods that update a subset of backbone parameters or add new trainable parameters to the backbone during fine-tuning:

- Sidetune (Zhang et al., 2020): train a "side" network and linearly interpolate between pre-trained features and side-tuned features before feeding them into the head.
- Bias (Cai et al., 2020; Zaken et al., 2022): fine-tune only the bias terms of a pre-trained backbone.
- Adapter (Houlsby et al., 2019; Pfeiffer et al., 2020a;b): insert new MLP modules with residual connections inside Transformer layers.
- LoRA (Hu et al., 2022; He et al., 2023; Fang et al., 2024): insert low-rank trainable matrices into the projection layers while freezing the original backbone weights, enabling parameter-efficient adaptation.

Table 8: FGVC per-task results for ViT-Base/16 pretrained on supervised ImageNet-21k.

| ViT-Base/16 | FGVC | | | | | Mean | Convergence |
| | CUB-200-2011 | NAbirds | Oxford Flowers | Stanford Dogs | Stanford Cars | | epoch |
|---|---|---|---|---|---|---|---|
| FULL | 87.3 | 82.7 | 98.8 | 89.4 | 84.5 | 88.54 | - |
| LINEAR | 85.3 | 75.9 | 97.9 | 86.2 | 51.3 | 79.32 | - |
| PARTIAL-1 | 85.6 | 77.8 | 98.2 | 85.5 | 66.2 | 82.63 | - |
| MLP-2 | 85.7 | 77.2 | 98.2 | 85.4 | 54.9 | 80.28 | - |
| MLP-3 | 85.1 | 77.3 | 97.9 | 84.9 | 53.8 | 79.80 | - |
| MLP-5 | 84.2 | 76.7 | 97.6 | 84.8 | 50.2 | 78.71 | - |
| MLP-9 | 83.2 | 76.0 | 96.2 | 83.7 | 47.6 | 77.31 | - |
| SIDETUNE | 84.7 | 75.8 | 96.9 | 85.8 | 48.6 | 78.35 | - |
| BIAS | 88.4 | 84.2 | 98.8 | 91.2 | 79.4 | 88.41 | - |
| ADAPTER-256 | 87.2 | 84.3 | 98.5 | 89.9 | 68.6 | 85.70 | - |
| ADAPTER-64 | 87.1 | 84.3 | 98.5 | 89.8 | 68.6 | 85.67 | - |
| ADAPTER-8 | 87.3 | 84.3 | 98.4 | 88.8 | 68.4 | 85.46 | - |
| LoRA | 88.3 | 85.6 | 99.2 | 91.0 | 83.2 | 89.46 | - |
| SPT-LoRA | 88.6 | 83.4 | 99.5 | 91.4 | 87.3 | 90.04 | - |
| DM-LoRA | 89.8 | 86.6 | 99.5 | 91.8 | 85.7 | 90.68 | - |
| VPT | 88.5 | 84.2 | 99.0 | 90.2 | 83.6 | 89.11 | 92 |
| VPT + PAE | 89.7 | 87.4 | 99.2 | 92.0 | 86.8 | 91.02 | 54 |
| E2VPT | 89.1 | 84.6 | 99.1 | 90.5 | 82.8 | 89.22 | 96 |
| E2VPT + PAE | 90.3 | 87.4 | 99.3 | 91.4 | 86.4 | 90.96 | 58 |
| LPT | 89.6 | 86.1 | 99.0 | 90.9 | 84.2 | 89.96 | 87 |
| LPT + PAE | 90.8 | 87.6 | 99.4 | 91.7 | 87.2 | 91.32 | 58 |
| VQT | 88.9 | 85.5 | 99.2 | 90.2 | 83.1 | 89.41 | 85 |
| VQT + PAE | 90.0 | 86.6 | 99.1 | 91.5 | 85.9 | 90.62 | 56 |
| VFPT | 88.7 | 84.5 | 99.1 | 90.4 | 83.6 | 89.24 | 76 |
| VFPT + PAE | 90.5 | 88.4 | 99.3 | 92.7 | 86.5 | 91.48 | 61 |
| SA2VP | 89.1 | 85.8 | 99.3 | 92.1 | 84.1 | 90.08 | 81 |
| SA2VP + PAE | 89.3 | 88.1 | 99.5 | 92.0 | 87.1 | 91.20 | 50 |
| ProVP | 89.6 | 84.9 | 99.0 | 90.5 | 83.8 | 89.56 | 84 |
| ProVP + PAE | 90.4 | 87.3 | 99.2 | 92.3 | 86.7 | 91.18 | 70 |
| BPT | 90.2 | 87.5 | 99.7 | 90.1 | 86.8 | 90.86 | 93 |
| BPT + PAE | 91.2 | 89.0 | 99.6 | 91.9 | 89.4 | 92.21 | 63 |

Table 9: VTAB-1k *Natural* per-task results for ViT-Base/16 pretrained on supervised ImageNet-21k.

| ViT-Base/16 | VTAB-1k *Natural* | | | | | | | Mean | Convergence |
| | CIFAR-100 | Caltech101 | DTD | Flowers102 | Pets | SVHN | Sun397 | | epoch |
|---|---|---|---|---|---|---|---|---|---|
| FULL | 68.9 | 87.7 | 64.3 | 97.2 | 86.9 | 87.4 | 38.8 | 75.88 | - |
| LINEAR | 63.4 | 85.0 | 63.2 | 97.0 | 86.3 | 36.6 | 51.0 | 68.93 | - |
| PARTIAL-1 | 66.8 | 85.9 | 62.5 | 97.3 | 85.5 | 37.6 | 50.6 | 69.44 | - |
| MLP-2 | 63.2 | 84.8 | 60.5 | 97.6 | 85.9 | 34.1 | 47.8 | 67.70 | - |
| MLP-3 | 63.8 | 84.7 | 62.3 | 97.4 | 84.7 | 32.5 | 49.2 | 67.80 | - |
| MLP-5 | 59.3 | 84.4 | 59.9 | 96.1 | 84.4 | 30.9 | 46.8 | 65.98 | - |
| MLP-9 | 53.1 | 80.5 | 53.9 | 95.1 | 82.6 | 24.4 | 43.7 | 61.90 | - |
| SIDETUNE | 60.7 | 60.8 | 53.6 | 95.5 | 66.7 | 34.9 | 35.3 | 58.21 | - |
| BIAS | 72.8 | 87.0 | 59.2 | 97.5 | 85.3 | 59.9 | 51.4 | 73.30 | - |
| ADAPTER-256 | 74.1 | 86.1 | 63.2 | 97.7 | 87.0 | 34.6 | 50.8 | 70.50 | - |
| ADAPTER-64 | 74.2 | 85.8 | 62.7 | 97.6 | 87.2 | 36.3 | 50.9 | 70.65 | - |
| ADAPTER-8 | 74.2 | 85.7 | 62.7 | 97.8 | 87.2 | 36.4 | 50.7 | 70.67 | - |
| LoRA | 67.1 | 91.4 | 69.4 | 98.8 | 90.4 | 85.3 | 54.0 | 79.46 | - |
| SPT-LoRA | 73.5 | 93.3 | 72.5 | 99.3 | 91.5 | 87.9 | 55.5 | 81.93 | - |
| DM-LoRA | 74.0 | 90.7 | 73.9 | 99.3 | 92.2 | 91.1 | 56.4 | 82.51 | - |
| VPT | 78.8 | 90.8 | 65.8 | 98.0 | 88.3 | 78.1 | 49.6 | 78.48 | 81 |
| VPT + PAE | 79.5 | 92.3 | 70.4 | 98.6 | 91.7 | 88.2 | 51.4 | 81.73 | 36 |
| E2VPT | 78.6 | 89.4 | 67.8 | 98.2 | 88.5 | 85.3 | 52.3 | 80.01 | 91 |
| E2VPT + PAE | 79.1 | 91.3 | 69.4 | 98.0 | 89.8 | 87.6 | 54.5 | 81.39 | 59 |
| LPT | 77.2 | 89.9 | 66.4 | 97.8 | 89.1 | 83.4 | 50.9 | 79.24 | 88 |
| LPT + PAE | 78.8 | 91.5 | 69.7 | 98.0 | 91.3 | 86.1 | 51.7 | 81.01 | 61 |
| VQT | 77.9 | 88.4 | 68.2 | 98.1 | 89.6 | 82.8 | 51.2 | 79.46 | 80 |
| VQT + PAE | 80.3 | 91.0 | 73.6 | 99.2 | 92.5 | 85.7 | 54.4 | 82.39 | 52 |
| VFPT | 80.7 | 91.4 | 69.4 | 99.3 | 90.3 | 85.6 | 52.7 | 81.35 | 77 |
| VFPT + PAE | 82.1 | 91.8 | 69.5 | 99.6 | 91.7 | 87.2 | 52.6 | 82.07 | 62 |
| SA2VP | 73.0 | 91.9 | 70.5 | 99.1 | 90.8 | 84.7 | 56.8 | 80.97 | 86 |
| SA2VP + PAE | 77.5 | 93.2 | 70.1 | 99.6 | 91.3 | 87.8 | 56.5 | 82.86 | 54 |
| ProVP | 75.2 | 91.7 | 68.7 | 99.0 | 89.8 | 84.5 | 53.6 | 80.35 | 89 |
| ProVP + PAE | 80.6 | 92.8 | 71.3 | 99.4 | 89.3 | 86.7 | 56.2 | 82.33 | 74 |
| BPT | 74.6 | 90.8 | 69.4 | 99.5 | 90.2 | 84.5 | 52.7 | 80.24 | 79 |
| BPT + PAE | 79.3 | 93.5 | 70.8 | 99.5 | 92.7 | 86.1 | 55.3 | 82.46 | 61 |

Table 10: VTAB-1k *Specialized* per-task results for ViT-Base/16 pretrained on supervised ImageNet-21k.

| ViT-Base/16 | VTAB-1k *Specialized* (4) | | | | Mean | Convergence epoch |
|---|---|---|---|---|---|---|
| | Patch Camelyon | EuroSAT | Resisc45 | Retinopathy | | |
| FULL | 79.7 | 95.7 | 84.2 | 73.9 | 83.36 | - |
| LINEAR | 78.5 | 87.5 | 68.6 | 74.0 | 77.16 | - |
| PARTIAL-1 | 78.6 | 89.8 | 72.5 | 73.3 | 78.53 | - |
| MLP-2 | 74.3 | 88.8 | 67.1 | 73.2 | 75.86 | - |
| MLP-3 | 77.0 | 88.0 | 70.2 | 56.1 | 72.83 | - |
| MLP-5 | 73.7 | 87.2 | 64.8 | 71.5 | 74.31 | - |
| MLP-9 | 78.5 | 83.0 | 60.2 | 72.3 | 73.49 | - |
| SIDETUNE | 58.5 | 87.7 | 65.2 | 61.0 | 68.12 | - |
| BIAS | 78.7 | 91.6 | 72.9 | 69.8 | 78.25 | - |
| ADAPTER-256 | 76.3 | 88.0 | 73.1 | 70.5 | 76.98 | - |
| ADAPTER-64 | 76.3 | 87.5 | 73.7 | 70.9 | 77.10 | - |
| ADAPTER-8 | 76.9 | 89.2 | 73.5 | 71.6 | 77.80 | - |
| LoRA | 84.9 | 95.3 | 84.4 | 73.6 | 84.55 | - |
| SPT-LoRA | 85.7 | 96.2 | 85.9 | 75.9 | 85.93 | - |
| DM-LoRA | 85.6 | 96.5 | 87.0 | 76.1 | 86.30 | - |
| VPT | 81.8 | 96.1 | 83.4 | 68.4 | 82.43 | 69 |
| VPT + PAE | 83.1 | 96.8 | 85.7 | 72.5 | 84.52 | 34 |
| E2VPT | 82.5 | 96.8 | 84.8 | 73.6 | 84.43 | 90 |
| E2VPT + PAE | 83.6 | 97.4 | 86.1 | 75.9 | 85.76 | 56 |
| LPT | 81.6 | 95.5 | 84.7 | 71.8 | 83.40 | 83 |
| LPT + PAE | 82.8 | 96.4 | 85.3 | 75.6 | 85.02 | 55 |
| VQT | 81.4 | 94.8 | 84.2 | 72.5 | 82.23 | 78 |
| VQT + PAE | 83.5 | 96.1 | 86.8 | 74.9 | 85.34 | 51 |
| VFPT | 83.5 | 96.5 | 84.4 | 75.4 | 84.93 | 72 |
| VFPT + PAE | 83.4 | 97.8 | 85.9 | 76.7 | 85.96 | 58 |
| SA2VP | 86.0 | 95.9 | 85.8 | 75.2 | 85.73 | 81 |
| SA2VP + PAE | 85.7 | 97.5 | 86.9 | 76.2 | 86.58 | 53 |
| ProVP | 84.6 | 95.7 | 82.4 | 73.6 | 84.07 | 87 |
| ProVP + PAE | 84.8 | 96.1 | 85.0 | 75.7 | 85.41 | 73 |
| BPT | 85.8 | 96.2 | 81.5 | 74.9 | 84.45 | 88 |
| BPT + PAE | 87.4 | 97.6 | 84.1 | 76.2 | 86.33 | 60 |

Table 11: VTAB-1k *Structured* per-task results for ViT-Base/16 pretrained on supervised ImageNet-21k.

| ViT-Base/16 (85.8M) | VTAB-1k *Structured* | | | | | | | | Mean | Convergence epoch |
|---|---|---|---|---|---|---|---|---|---|---|
| | Clevr/ count | Clevr/ distance | DMLab | KITTI/ distance | dSprites/ location | dSprites/ orientation | SmallNORB/ azimuth | SmallNORB/ elevation | | |
| FULL | 56.3 | 58.6 | 41.7 | 65.5 | 57.5 | 46.7 | 25.7 | 29.1 | 47.64 | - |
| LINEAR | 34.3 | 30.6 | 33.2 | 55.4 | 12.5 | 20.0 | 9.6 | 19.2 | 26.84 | - |
| PARTIAL-1 | 41.5 | 34.3 | 33.9 | 61.0 | 31.3 | 32.8 | 16.3 | 22.4 | 34.17 | - |
| MLP-2 | 45.2 | 31.6 | 31.8 | 55.7 | 30.9 | 24.6 | 16.6 | 23.3 | 32.47 | - |
| MLP-3 | 47.8 | 32.8 | 32.3 | 58.1 | 12.9 | 21.2 | 15.2 | 24.8 | 30.62 | - |
| MLP-5 | 50.8 | 32.3 | 31.5 | 56.4 | 7.5 | 20.8 | 14.4 | 20.4 | 29.23 | - |
| MLP-9 | 47.5 | 27.9 | 28.9 | 54.0 | 6.2 | 17.7 | 10.8 | 16.2 | 26.15 | - |
| SIDETUNE | 27.6 | 22.6 | 31.3 | 51.7 | 8.2 | 14.4 | 9.8 | 21.8 | 23.41 | - |
| BIAS | 61.5 | 55.6 | 32.4 | 55.9 | 66.6 | 40.0 | 15.7 | 25.1 | 44.09 | - |
| ADAPTER-256 | 45.7 | 37.4 | 31.2 | 53.2 | 30.3 | 25.4 | 13.8 | 22.1 | 32.39 | - |
| ADAPTER-64 | 42.9 | 39.9 | 30.4 | 54.5 | 31.9 | 25.6 | 13.5 | 21.4 | 32.51 | - |
| ADAPTER-8 | 45.2 | 41.8 | 31.1 | 56.4 | 30.4 | 24.6 | 13.2 | 22.0 | 33.09 | - |
| LoRA | 82.9 | 69.2 | 49.8 | 78.5 | 75.7 | 47.1 | 31.0 | 44.4 | 59.83 | - |
| T-LoRA | 84.4 | 67.6 | 52.5 | 82.0 | 81.0 | 51.1 | 30.2 | 41.3 | 61.26 | - |
| DM-LoRA | 83.5 | 69.9 | 52.0 | 81.6 | 80.2 | 50.2 | 36.1 | 43.1 | 62.08 | - |
| VPT | 68.5 | 60.0 | 46.5 | 72.8 | 73.6 | 47.9 | 32.9 | 37.8 | 54.98 | 78 |
| VPT + PAE | 73.4 | 61.7 | 46.9 | 75.8 | 77.9 | 53.6 | 34.3 | 42.6 | 58.28 | 56 |
| E2VPT | 71.7 | 61.2 | 47.9 | 75.8 | 80.8 | 48.1 | 31.7 | 41.9 | 57.39 | 95 |
| E2VPT + PAE | 72.9 | 63.7 | 49.4 | 78.6 | 82.5 | 52.9 | 33.8 | 44.3 | 59.73 | 60 |
| LPT | 70.6 | 61.8 | 47.1 | 73.6 | 78.8 | 51.6 | 33.2 | 42.4 | 57.39 | 82 |
| LPT + PAE | 74.8 | 63.5 | 49.6 | 75.8 | 80.6 | 54.7 | 32.9 | 43.4 | 59.41 | 60 |
| VQT | 71.8 | 62.6 | 46.4 | 73.7 | 79.3 | 50.8 | 32.5 | 42.6 | 57.47 | 83 |
| VQT + PAE | 75.1 | 64.4 | 48.8 | 77.2 | 81.6 | 52.6 | 33.8 | 45.1 | 59.83 | 55 |
| VFPT | 75.8 | 63.2 | 48.3 | 79.3 | 81.5 | 56.0 | 34.1 | 43.4 | 60.19 | 74 |
| VFPT + PAE | 76.9 | 65.2 | 48.1 | 79.7 | 82.4 | 55.2 | 36.5 | 43.7 | 60.96 | 56 |
| SA2VP | 76.6 | 71.8 | 50.8 | 79.9 | 84.5 | 52.8 | 34.7 | 45.3 | 60.80 | 84 |
| SA2VP + PAE | 78.2 | 72.5 | 53.3 | 79.6 | 86.4 | 54.5 | 34.1 | 45.8 | 63.05 | 51 |
| ProVP | 75.5 | 67.2 | 49.3 | 77.6 | 83.7 | 51.9 | 33.8 | 43.5 | 60.31 | 84 |
| ProVP + PAE | 77.3 | 71.6 | 51.5 | 76.8 | 83.5 | 53.6 | 34.7 | 44.7 | 61.72 | 71 |
| BPT | 76.1 | 65.3 | 48.9 | 77.8 | 82.5 | 53.2 | 34.7 | 44.6 | 60.39 | 80 |
| BPT + PAE | 77.6 | 70.1 | 52.5 | 77.2 | 84.7 | 53.9 | 35.1 | 45.3 | 62.05 | 64 |

