# OpenReview forum: "Visual Prompt-Agnostic Evolution"
_ICLR.cc/2026/Conference — ICLR 2026 Poster_

### Official Review · Reviewer_oUNy · 2025-10-14

**Soundness:** 2
**Presentation:** 2
**Contribution:** 2
**Rating:** 4
**Confidence:** 3

**Summary:**

Visual cue adaptation effectively adapts frozen pre-trained models by inserting a small number of learnable cue tokens into each layer of ViT. However, existing VPT variants often suffer from dynamic instability in training, manifested by gradient oscillations, early stagnation of shallow-layer cues, and high-variance oscillations of deep-layer cues, leading to slow convergence and ultimately degraded performance. To address these challenges, this paper proposes Prompt-Agnostic Evolution (PAE), which enhances VPT by explicitly modeling the dynamic evolution of learnable cues.

**Strengths:**

1. PAE is lightweight and independent of prompts, a significant decoupling advantage.
2. It performs well, especially on classification tasks.
3. The derivation is robust, with no major issues.

**Weaknesses:**

This method introduces multiple new hyperparameters, increasing the complexity of model tuning. Ablation experiments in the paper show that model performance is sensitive to the choice of these parameters. This means that in real-world applications, tedious search and fine-tuning for different tasks may be required to achieve optimal results, which somewhat limits its plug-and-play usability.

The experimental validation in the paper focuses primarily on the specific architecture ViT-Base and the task of image classification. While achieving convincing results on the FGVC and VTAB-1k datasets, the effectiveness of this method on a wider range of visual tasks, such as object detection and semantic segmentation, has not yet been verified. This raises questions about its general applicability.

**Questions:**

The paper simplifies the evolution of cues across layers to a global linear transformation (the Koopman operator) in a shared latent space. While this linear assumption effectively promotes inter-layer consistency and stabilizes training, does it limit the model's expressive power, especially when dealing with complex tasks or very deep networks with highly nonlinear inter-layer relationships? Exploring nonlinear dynamical models or introducing layer-wise adaptive evolutionary operators might yield further improvements in model performance at the expense of a small degree of simplicity.

Reframing cue tuning as a dynamical system is a novel perspective. In addition to stabilizing training and accelerating convergence, can this framework also be used to improve model interpretability? For example, by analyzing the eigenvalues ​​and eigenvectors of the learned Koopman operator, can we gain insight into how the model adjusts its internal representation to adapt to new tasks? Or can we identify the "dynamic patterns" that are crucial for specific tasks, thereby providing theoretical guidance for more effective cue design?

---

> ### Author Response · Authors · 2025-11-21
> **Response to Reviewer oUNy (1/2)**
>
> W1: Hyperparameter sensitivity.
>
> We thank the reviewer for raising this concern. In practice, each hyperparameter in our design has a clear role and a straightforward selection rule, allowing users to rely on a small set of robust defaults rather than performing task-specific tuning.
>
> 1. **Frequency parameters $ (w, r) $:**  Inspired by the multi-round frequency–shortcut analysis of ImageNet models in [1], we simplify their procedure to a **single-round sweep**: (i) restrict $ (w, r) $ to a low–to–mid frequency band that covers the shortcut-prone ranges reported in [1], then (ii) perform a small grid search within this band once to pick a working setting. Our goal is to define a *sensible range* rather than a closed-form optimum, and Table 6 shows that accuracy is very stable (typically $<0.8\%$ fluctuation) across this entire band—any choice inside it is generally good.
> 2. **Loss weights $ (\alpha, \beta) $:**  For $ (\alpha, \beta) $, we follow standard practice for constraint and regularization terms: start from balanced weights, then rescale so that the Koopman and stability losses have gradient magnitudes comparable to the base VPT classification loss. Table 7 confirms that performance is flat over a reasonably wide interval, again avoiding fine-grained tuning.
> 3. **Koopman dimension $ K $:**  $ K $ is introduced to realize the **first application of Koopman theory to visual prompt tuning**, rather than as an arbitrary extra knob. Fig. 6 shows a broad plateau for $ K \in [128, 320] $ across all VTAB-1k groups; we therefore fix $ K = 256 $ as a *single* default for **all** 25 datasets and architectures (ViT, Swin, SETR), without any per-task retuning.
>
> **Summary:**  Although PAE formally introduces several hyperparameters, each comes with a principled design motivation and a broad, empirically validated stable range. This allows us to use a single shared configuration across 25 datasets and 5 architectures. In practice, users can either adopt our defaults or pick values within the documented bands—eliminating the need for the tedious task-specific search the reviewer is concerned about.
>
> [1] Do ImageNet-trained models learn shortcuts? The impact of frequency shortcuts on generalization. CVPR 2025.
>
>
>
> W2: Limited Experimental Scope.
>
> To demonstrate general applicability, we added a Semantic Segmentation experiment on ADE20K (Table 2).
>
> | Methods     | Speedup($\times$) |     mIoU-SS      |     mIoU-Ms      |
> | :---------- | :---------------: | :--------------: | :--------------: |
> | Full-tuning |         -         |      47.60       |      49.18       |
> | SPT-LoRA    |         -         |      45.40       |      47.50       |
> | VPT         |   1.29$\times$    | 44.08 + **2.73** | 46.01 + **1.96** |
> | E2VPT       |   1.18$\times$    | 44.61 + **2.32** | 46.56 + **2.84** |
> | VFPT        |   1.15$\times$    | 45.32 + **2.75** | 47.17 + **2.09** |
>
> 1. **Dense Prediction Task:** Applying $\mathtt{PAE}$ to ViT-L (SETR architecture) on ADE20K yields a **+2.73% mIoU** improvement over standard VPT, proving effectiveness on pixel-level tasks.
> 2. **Scalability (ViT-B $\to$ ViT-H):** As shown in **Figure 6**, we validated $\mathtt{PAE}$ across varying model scales (ViT-B, ViT-L, ViT-H) and diverse architectures (Swin Transformer). $\mathtt{PAE}$ consistently improves performance across all settings.
> 3. **Self-Supervised Paradigm (MAE):** We further verified $\mathtt{PAE}$ on a Masked Autoencoder (MAE) backbone. As analyzed in **Figure 7**, $\mathtt{PAE}$ successfully establishes a structured, diagonal evolutionary trajectory on self-supervised features, demonstrating robustness across different pre-training paradigms.

---

> > ### Author Response · Authors · 2025-11-21
> > **Response to Reviewer oUNy (2/2)**
> >
> > Q1: Does the linear operator limit expressiveness?
> >
> > We have verified $\mathtt{PAE}$ on very deep networks (ViT-H, 32 layers) and complex dense prediction tasks (ADE20K), proving that our shared linear operator is sufficient to handle high non-linearity.
> >
> > To directly address your concern about expressiveness, we compared our approach against a **"Layer-wise Adaptive Design."** In this variant, instead of a single shared operator $\mathbf{K}$, we assign a distinct, independent learnable operator $\mathbf{K}_i$ to each layer transition ($i \to i+1$) to maximize theoretical expressiveness.
> >
> > However, our empirical results favor the shared global design for three reasons:
> >
> > 1. **Resource Efficiency:** A layer-wise design requires learning $(L-1)$ distinct operators instead of just one (global). For the ViT-B, this increases the computational resources for the dynamics module by roughly **10x** (e.g., 11 separate operators vs. one shared operator in ViT-B), contradicting the core principle of Parameter-Efficient Fine-Tuning (PEFT).
> > 2. **Stability Findings:** As shown in Fig. 10, layer-wise operators empirically tend to learn unstable modes (eigenvalues > 1), leading to exploding feature magnitudes. This instability degrades performance. In VTAB-1k benchmark, the layer-wise variant achieves **73.39%** mean accuracy, being **below** our stable shared operator (**74.84%**).
> > 3. **Why Linear evolution works:** The backbone itself is highly non-linear. The prompt's role is to guide this backbone. A stable, linear evolution of the prompt acts as a smooth control signal, which empirically yields better results than a complex, unstable prompt trajectory.
> >
> >
> >
> > Q2: Interpretability of the dynamic system.
> >
> > The dynamical perspective offers unique insights unavailable in VPT. Beyond verifying general stability (Fig. 1) and depth-wise consistency (Fig. 7), the spectrum of the Koopman operator $\mathbf{K}$ in Fig. 10 reveals **interpretable, dynamic patterns**:
> >
> > 1. **"Mild Reshaping" for Natural Tasks:** For standard natural images (e.g., CIFAR, Caltech), the operator learns a **moderate spectral radius** ($|\lambda| \approx 0.58$). This indicates that since the domain gap from the pre-trained ImageNet backbone is small, $\mathtt{PAE}$ only needs to gently adjust the feature trajectory to align with the new task.
> > 2. **"Long-Term Memory" for Specialized Tasks:** In domains with significant shifts like medical or satellite imagery, the operator exhibits the **largest spectral radius** ($|\lambda| \approx 0.67$) with modes persisting near the unit circle. This interpretable pattern suggests a strategy of **retaining information** strongly across layers to counteract the large distribution shift.
> > 3. **"Aggressive Damping" for Structured Tasks:** Conversely, for synthetic or geometric tasks (e.g., dSprites), the spectrum shows the **strongest contraction** ($|\lambda| \approx 0.44$). This reveals that the model actively **suppresses** the texture-heavy features (an inherent bias of the backbone) to focus on the geometric or rule-based signals required by these tasks.
> >
> > **Conclusion:** These spectral signatures prove that $\mathtt{PAE}$ learns an **adaptive evolutionary law** tailored to the specific semantic distance between the source and target domains, offering a level of interpretability that static prompt tuning cannot provide.

---

> ### Comment · Reviewer_oUNy · 2025-11-21
>
> I think author has addressed my concerns, I will improve my scores

---

> > ### Author Response · Authors · 2025-11-25
> >
> > We sincerely appreciate your feedback and support for our work. Please accept our apologies for the delay in this response.

---

### Official Review · Reviewer_9KV8 · 2025-10-18

**Soundness:** 3
**Presentation:** 3
**Contribution:** 3
**Rating:** 8
**Confidence:** 4

**Summary:**

Visual prompt tuning enables parameter-efficient fine-tuning. However, existing VPT variants often suffer from unstable training. To address this challenge, the authors propose the Prompt-Agnostic Evolution (PAE) by explicitly modeling the dynamics of learnable prompts.

**Strengths:**

1. The problem of unstable VPT training is well defined. The observations on shallow- and deep-layer prompts are interesting. The clear mismatch for gradient oscillations is something researchers might be interested in.

2. The paper is easy to follow, and the problem-solving is practical.

3. The ablation study is sufficient.

**Weaknesses:**

1. The masking then project idea sounds similar to projection-based (a.k.a instance-aware) prompt tuning [1-2], where these papers use input projection directly to guide prompt training. The authors need to discuss them and clearly separate their differences.

2. The format in conclusion is a little bit weird. Please fix it.

3. The motivation of this paper can be clearer, for example, why the authors want to discover frequency shortcuts. I understand the observations; however, their motivations are unclear to me.

[1] Visual Instance-aware Prompt Tuning

[2] All You Need is One: Capsule Prompt Tuning with a Single Vector

**Questions:**

My question mainly focuses on the discussions on projection-based prompt tuning methods. Other than that, this paper looks good to me.

---

> ### Author Response · Authors · 2025-11-21
> **Response to Reviewer 9KV8**
>
> W1/Q1: Comparison with projection-based methods.
>
> We appreciate this point. The key distinction lies in **inference efficiency**:
>
> 1. **Projection-based [1] [2]:** These methods keep an instance-aware projection in the loop throughout training and inference, tightly coupling the projection with prompt optimization, increasing computational cost.
> 2. **Ours ($\mathtt{PAE}$):** MPA is a **decoupled pre-alignment** step. We use the projection only once to initialize the prompts. During training and inference, our prompts are static vectors. This ensures our method maintains the zero-overhead advantage of standard VPT. We have clarified this distinction in the introduction section.
>
> [1] Visual Instance-aware Prompt Tuning
>
> [2] All You Need is One: Capsule Prompt Tuning with a Single Vector
>
>
>
>
> W2: Formatting in the Conclusion.
>
> We thank the reviewer for spotting this. We have corrected the formatting issues in the **Conclusion** section to ensuring a clean and professional presentation in the final version.
>
>
>
> W3: Clarity of Motivation.
>
> We appreciate the reviewer's suggestion to clarify our motivation. Our approach addresses two critical issues in existing VPT variants: **Optimization Instability** and **Task-Agnostic Initialization**.
>
> 1. Addressing Optimization Instability via Dynamical Systems ($\mathtt{KLD}$):
>
>    Standard VPT suffers from a "cross-layer mismatch": shallow prompts stagnate while deep prompts oscillate (Fig. 1c). Simple smoothing (like EMA) fails because it causes gradients to collapse to near-zero ($10^{-3}$), leading to slow convergence and suboptimal accuracy (Fig. 1a).
>
>    - **Our Solution:** We introduce a Koopman-Lyapunov Dynamical System ($\mathtt{KLD}$) not merely to "smooth" the trajectory, but to enforce a **controlled, coherent evolution**. By learning a stable operator $K$ ($\rho < 1$), $\mathtt{PAE}$ ensures that prompts evolve progressively across layers without exploding or vanishing, maintaining healthy gradient magnitudes ($10^{-1}$) and accelerating convergence.
>
> 2. Resolving Task-Agnostic Initialization via Frequency Shortcuts ($\mathtt{MPA}$):
>
>    Existing methods initialize prompts randomly (or with ImageNet pre-trained weights), which are task-agnostic. The model must waste early epochs "aligning" these random prompts to the downstream task, causing initial loss spikes and instability.
>
>    - **Why Frequency Shortcuts?** Recent studies [1,2] reveal that Vision Transformers heavily rely on specific frequency bands (e.g., high-frequency textures or low-frequency shapes) to make predictions. These "frequency shortcuts" are the most discriminative, task-specific signals inherent in the backbone.
>    - **Our Solution:** MPA explicitly identifies these discriminative frequency components *before* training and injects them into the initial prompts. This provides a **task-aware "hot start"**, effectively skipping the alignment phase and directly launching optimization in a task-relevant subspace.
>
> [1] What do neural networks learn in image classification? A frequency shortcut perspective. ICCV 2023.
>
> [2] Do ImageNet-trained models learn shortcuts? The impact of frequency shortcuts on generalization. CVPR 2025.

---

> > ### Comment · Reviewer_9KV8 · 2025-11-25
> >
> > Thank you for the clarification. I really like this work and thus have no further questions. I will maintain my current ratings.

---

> > > ### Author Response · Authors · 2025-11-25
> > >
> > > We are thrilled to hear that you enjoyed our work. We sincerely appreciate your strong approval.

---

### Official Review · Reviewer_PEzU · 2025-10-24

**Soundness:** 3
**Presentation:** 3
**Contribution:** 3
**Rating:** 6
**Confidence:** 3

**Summary:**

This paper addresses a core instability issue in Visual Prompt Tuning (VPT), a fine-tuning method for Vision Transformers (ViTs). The authors observe that many VPT variants suffer from unstable training processes, characterized by gradient oscillations and a cross-layer mismatch phenomenon, where prompts in shallower layers stagnate while those in deeper layers oscillate to compensate. This ultimately slows down convergence and hinders optimal performance. The paper identifies two root causes for this problem: task-agnostic prompt initialization and the independent, uncooperative optimization of prompts at each layer. To resolve this, the authors propose Prompt-Agnostic Evolution (PAE), a framework that treats prompt tuning as a dynamical system. PAE consists of two novel components. The first is Modal Pre-Alignment (MPA), which provides a task-aware initialization by identifying the most discriminative frequency shortcuts for a given task and using them to generate initial prompts. The second is the Koopman-Lyapunov Discrete Dynamical System (KLD), which governs the prompt optimization. It uses a shared Koopman operator to enforce a coherent linear evolution for prompts across multiple layers within a shared latent space, and a Lyapunov-style regularizer to ensure this evolution remains stable. In experiments on the FGVC and VTAB-1k benchmarks, applying PAE to various VPT methods consistently improved accuracy by 1-3% and accelerated convergence by an average of 1.48x. As a prompt-agnostic module, PAE can be integrated into existing methods without modifying the model's backbone and has zero overhead at inference time, making it a practical and effective solution.

**Strengths:**

The primary strength of this research is its novel conceptualization of prompt tuning as a dynamical system, providing a principled framework to address the observed training instabilities. By applying the Koopman operator and Lyapunov stability theory, it moves beyond empirical heuristics and introduces an explicit mechanism to coordinate prompt updates across layers, directly tackling the optimization mismatch problem. Another key innovation is the Modal Pre-Alignment (MPA) strategy. This method effectively solves the cold start problem in prompt tuning by using a task-aware initialization based on frequency-domain analysis. By identifying the frequency shortcuts already utilized by the pre-trained backbone, MPA provides initial prompts that are well-aligned with the task objective from the outset, which emerged as the single largest contributor to performance gains. Finally, the PAE framework demonstrates remarkable robustness and practicality. Its prompt-agnostic design allows it to be seamlessly integrated as a plug-and-play module into numerous state-of-the-art VPT variants, consistently improving performance in all cases. This proves that PAE addresses a fundamental weakness in the VPT paradigm. The comprehensive empirical validation, which includes not only accuracy metrics but also insightful loss landscape and Grad-CAM visualizations, provides strong evidence for its effectiveness. The fact that there is zero inference-time overhead further solidifies its value for real-world applications.

**Weaknesses:**

Assumptions of the KLD Framework: The Koopman-Lyapunov Discrete Dynamical System (KLD) assumes that the prompt dynamics can be effectively modeled by a single, global, linear operator. This could create a representational bottleneck for complex tasks where different dynamics in shallow and deep layers might be more beneficial. The framework's success also hinges on the assumption that prompt evolution is approximately linear in the learned latent space, which has not been validated across diverse model architectures and scales. The Lyapunov-style stability constraint, while effective, might be overly restrictive, potentially preventing the model from exploring optimal solutions that require a temporary increase in complexity. Consequently, although performance has been demonstrated, these are results from a single model, and the effectiveness of these simplifications (a single global operator and regularization) may diminish as conditions become more complex.

Limited Experimental Scope: The paper's empirical validation is limited to a single backbone architecture (ViT-Base/16), so it remains unverified whether PAE's effectiveness generalizes to other architectures. It also does not investigate how performance varies with model scale (e.g., larger or smaller ViTs). Lastly, the absence of a direct comparative analysis with other major PEFT (Parameter-Efficient Fine-Tuning) families, such as LoRA, makes it difficult to fully assess the pros and cons of PAE-enhanced VPT within the broader PEFT landscape. An exploration of its orthogonality and complementarity with these methods would be beneficial.

In conclusion, the study has demonstrated the success of the KLD framework's simplifying assumptions within the specific context of ViT-Base/16. However, it has not been verified whether this success will hold under more complex conditions, such as with more intricate architectures, much larger-scale models, or when combined with other PEFT techniques like LoRA. In other words, a key limitation of this research is that the possibility that the success of this simplification is a coincidence within the limited experimental scope cannot be ruled out.

**Questions:**

I would appreciate your response to the points raised in the "Weakness" section.

Additionally:
- I would like to know how the convergence speed varies with changes in the hyperparameters alpha and beta.
- I am interested in the performance differences based on intra-class variance. It would be helpful to see the difference in performance gains between the best- and worst-performing classes or class groups in the dataset.

**Details Of Ethics Concerns:**

No ethic concerns

---

> ### Author Response · Authors · 2025-11-21
> **Response to Reviewer PEzU (1/3)**
>
> W1: Assumptions of the KLD Framework.
>
> We appreciate this comprehensive comment. We address the concerns regarding the theoretical assumptions of the KLD framework first, followed by our expanded experimental evaluation on larger models and diverse architectures.
>
> 1. **Validity of KLD Assumptions (Linearity & Global Operator)**
>
>    - **Theoretical Basis (Linearity via Lifting):** KLD does *not* assume that the raw prompt \(P\) follows linear dynamics. Instead, Koopman theory provides a mechanism to **lift** prompts into an observable space \(z = P U\) where the evolution becomes *approximately linear*. The learned projection \(U\) provides the expressiveness needed to capture non-linear depth trajectories, while the linear operator \(K\) enables stable and analyzable evolution. This avoids the representational bottleneck.
>
>    - **Why a Single Global Operator (Stability & Coherence):** To test whether different dynamics in shallow and deep layers might be beneficial, we implemented a **Layer-wise Adaptive Design** with independent operators \(K_l\) per layer. As shown in **Fig. 10(b–c)**, layer-wise operators frequently learn **unstable modes** \((|\lambda|>1)\), leading to exploding or chaotic prompt trajectories and degraded generalization. In contrast, a **single global operator** learns a stable spectrum \((\rho(K)<1)\), enforcing coherent cross-layer evolution and yielding significantly better convergence and robustness.
>
>    - **Evidence from Complex Tasks:** We also evaluate on **Semantic Segmentation (ADE20K)** using the **SETR** architecture (**Table 2**). The observed **+2.73% mIoU** improvement demonstrates that a single global operator is sufficiently expressive even for dense, high-complexity prediction tasks, not only classification.
>
> 2. **Expanded Evaluation: Large-scale Models & Diverse Architectures**
>
>    To further address concerns about expressiveness and generality, we expanded our evaluation across model scales and architectures (**Fig. 6, 7**):
>
>    - **Large-scale Models (Depth & Width).**
>      We added experiments on **ViT-L/16** and **ViT-H/14**, extending to 32-layer networks. Across all scales—from Base to Huge—$\mathtt{PAE}$ consistently improves over baselines, indicating that the global linear evolution assumption remains valid for deep, high-capacity models.
>
>    - **Architectural Diversity (Hierarchy & Pretraining).**
>      We further validate \($\mathtt{PAE}$\) on **Swin Transformer** (hierarchical, windowed attention). The consistent gains across such structurally different architectures demonstrate that the KLD formulation is not tied to ViTs and generalizes broadly.

---

> > ### Author Response · Authors · 2025-11-21
> > **Response to Reviewer PEzU (2/3)**
> >
> > W2: Limited Experimental Scope
> >
> > Please refer also to our response to **W1** for the expanded evaluation on large-scale models and diverse architectures. Here, we further provide comparison with LoRA [1], SPT-LoRA [2] and DM-LoRA [3] on FGVC and VTAB, as shown in the newly added tables. In the revision, we have incorporated these results into Table 8-11 of the appendix.
> >
> > (Note: Combining $\mathtt{PAE}$ with LoRA, e.g., applying Koopman dynamics to LoRA states, is conceptually feasible but requires additional design choices, so we leave it as promising future work.)
> >
> > FGVC
> >
> > | Method   | CUB-200-2011 | NAbirds | Oxford Flowers | Stanford Dogs | Stanford Cars | Mean  |
> > | -------- | ------------ | ------- | -------------- | ------------- | ------------- | ----- |
> > | LoRA     | 88.3         | 85.6    | 99.2           | 91.0          | 83.2          | 89.46 |
> > | SPT-LoRA | 88.6         | 83.4    | 99.5           | 91.4          | 87.3          | 90.04 |
> > | DM-LoRA  | 89.8         | 86.6    | 99.5           | 91.8          | 85.7          | 90.68 |
> >
> > VTAB-1k $\textit{Natural}$
> >
> > | Method   | CIFAR-100 | Caltech101 | DTD  | Flowers102 | Pets | SVHN | Sun397 | Mean  |
> > | -------- | --------- | ---------- | ---- | ---------- | ---- | ---- | ------ | ----- |
> > | LoRA     | 67.1      | 91.4       | 69.4 | 98.8       | 90.4 | 85.3 | 54.0   | 79.46 |
> > | SPT-LoRA | 73.5      | 93.3       | 72.5 | 99.3       | 91.5 | 87.9 | 55.5   | 81.93 |
> > | DM-LoRA  | 74.0      | 90.7       | 73.9 | 99.3       | 92.2 | 91.1 | 56.4   | 82.51 |
> >
> > VTAB-1k $\textit{Specialized}$
> >
> > | Method   | Patch Camelyon | EuroSAT | Resisc45 | Retinopathy | Mean  |
> > | -------- | -------------- | ------- | -------- | ----------- | ----- |
> > | LoRA     | 84.9           | 95.3    | 84.4     | 73.6        | 84.55 |
> > | SPT-LoRA | 85.7           | 96.2    | 85.9     | 75.9        | 85.93 |
> > | DM-LoRA  | 85.6           | 96.5    | 87.0     | 76.1        | 86.30 |
> >
> > VTAB-1k $\textit{Structured}$
> >
> > | Method  | Clevr (count) | Clevr (distance) | DMLab | KITTI (distance) | dSprites (location) | dSprites (orientation) | SmallNORB (azimuth) | SmallNORB (elevation) | Mean  |
> > | ------- | ------------- | ---------------- | ----- | ---------------- | ------------------- | ---------------------- | ------------------- | --------------------- | ----- |
> > | LoRA    | 82.9          | 69.2             | 49.8  | 78.5             | 75.7                | 47.1                   | 31.0                | 44.4                  | 59.83 |
> > | T-LoRA  | 84.4          | 67.6             | 52.5  | 82.0             | 81.0                | 51.1                   | 30.2                | 41.3                  | 61.26 |
> > | DM-LoRA | 83.5          | 69.9             | 52.0  | 81.6             | 80.2                | 50.2                   | 36.1                | 43.1                  | 62.08 |
> >
> > We believe the consistency of our results makes a mere coincidence unlikely:
> >
> > 1. **Consistency:** We tested on **25 different datasets** (FGVC + VTAB-1k + ADE20K) and **5 architectures** (ViT-B/L/H, Swin, SETR). The gains are consistent, not limited to specific tasks.
> >
> > 2. **Stability:** MPA with different random seeds (Table 5) show negligible variance, the results of paired t-tests are p ∈ [0.008, 0.092].
> >
> > 3. **Mechanism:** The spectral analysis (Fig. 10) supports that the model learns a stable evolution operator as intended, providing a mechanistic explanation for the performance.
> >
> > [1] Lora: Low-rank adaptation of large language models. ICLR 2022.
> >
> > [2] Sensitivity-aware visual parameter-efficient fine-tuning. CVPR 2024.
> >
> > [3] Dropout mixture low-rank adaptation for visual parameters-efficient fine-tuning. ECCV 2024.

---

> > > ### Author Response · Authors · 2025-11-21
> > > **Response to Reviewer PEzU (3/3)**
> > >
> > > Q1: How do hyperparameters ($\alpha$, $\beta$) affect convergence speed?
> > >
> > > We have analyzed the impact of varying $(\alpha, \beta)$ values on convergence dynamics. While MPA provides a "hot start," the values of $\alpha$ and $\beta$ determine the efficiency of the optimization trajectory. As illustrated in **Appendix Fig. 11,** the convergence speed follows a distinct U-shaped trend with respect to the regularization strength.
> > >
> > > 1. **Moderate weights (Optimal Region, e.g., $\alpha=0.5, \beta=0.2$):** This setting achieves the fastest convergence universally, requiring only **30–55 epochs** across tasks (blue/orange/green/red curves at the valley). By effectively suppressing the high-frequency gradient oscillations observed in baseline VPT, this stability allows for aggressive learning rates, driving the observed **1.41x speedup**.
> > >
> > >    **Too small (Under-regularization, e.g., $\alpha=0.1, \beta=0.1$):** Convergence slows significantly to **65–90 epochs**. Without sufficient constraints, the system reverts to uncoordinated layer-wise updates. Severe gradient oscillations reappear (similar to the baseline), forcing the optimization to traverse a jagged, inefficient path.
> > >
> > >    **Too large (Over-regularization, e.g., $\alpha=0.9, \beta=0.3$):** Convergence cost rebounds sharply to **70–95 epochs**. Strong regularization leads to "over-smoothing" of the prompt evolution. The model is overly constrained and struggles to capture task-specific discriminative features, resulting in underfitting (as evidenced by the accuracy drop in Table 7) and a prolonged struggle to reach a suboptimal plateau.
> > >
> > >
> > >
> > > Q2: Performance gains vs. Intra-class variance.
> > >
> > > We choose CUB-200-2011 dataset from the FGVC benchmark, which contains 200 fine-grained bird species with subtle inter-class differences but substantial intra-class variation. On CUB-200, we performed a detailed per-class breakdown in Fig. 8:
> > >
> > > 1. **High variance $\approx$ harder classes:** In Fig. 8(a), we measure the intra-class variance of VPT+$\mathtt{PAE}$ features and the corresponding per-class accuracy. We observe a mild **negative Pearson correlation** between variance and accuracy (corr = −0.290), which means that classes with higher intra-class variance tend to have lower accuracy. In other words, *class difficulty is positively correlated with intra-class variance*.
> > > 2. **Hard classes benefit more from $\mathtt{PAE}$:** In Fig. 8(b), we then plot the **accuracy gain of VPT+$\mathtt{PAE}$ over VPT** as a function of intra-class variance. Here we observe a **positive Pearson correlation** (corr = 0.207): higher-variance (harder) classes receive *larger relative improvements* from using $\mathtt{PAE}$ compared to low-variance (easier) classes.
> > > 3. **Conclusion:** These two trends show that $\mathtt{PAE}$ is *most effective exactly where standard VPT struggles*: on high-variance, hard classes, where stabilizing the optimization dynamics brings the largest gains.

---

> ### Comment · Reviewer_PEzU · 2025-11-26
>
> I thank the authors for their detailed and comprehensive response. The rebuttal has effectively addressed my primary concerns regarding the experimental scope and the theoretical assumptions of the KLD framework.
>
> Here are the key reasons for raising my score:
>
> **1. Comparison with PEFT Baselines:** The inclusion of comparative experiments with LoRA and its variants (SPT-LoRA, DM-LoRA) in the appendix is a significant addition. It clearly demonstrates that PAE-enhanced VPT is competitive with or superior to established PEFT methods, resolving my concern about the lack of relative performance assessment in the broader landscape.
>
> **2. Validation on Diverse Architectures and Scales:** Expanding the evaluation to larger models (ViT-L, ViT-H) and a different architecture (Swin Transformer) provides strong evidence that the proposed method generalizes well beyond the initial ViT-Base setup. This alleviates the concern that the results might be specific to a single model configuration.
>
> **3. Justification of the Global Operator Assumption:** The authors provided a compelling empirical justification for using a single global operator. The comparison with the Layer-wise Adaptive Design, which resulted in unstable modes and exploding trajectories, effectively supports the claim that the global linear evolution assumption is not merely a simplification but a necessary design choice for stability.
>
> **4. Insightful Analysis on Convergence and Difficulty:** The additional analysis on hyperparameter sensitivity and the correlation between intra-class variance and performance gains offers valuable insights. It highlights that PAE is particularly beneficial for hard classes with high variance, which adds depth to the contribution.
>
> The authors have strengthened the paper during the rebuttal phase by filling the experimental gaps and providing mechanistic evidence for their design choices. Given the promise of the VPT direction and the solid verification provided, I believe this paper makes a valuable contribution to ICLR. I am raising my score to 8.

---

> > ### Author Response · Authors · 2025-11-26
> >
> > We are delighted that our additional experiments and analyses effectively addressed your concerns. We deeply appreciate your constructive feedback, which has significantly strengthened the quality and depth of our work.

---

### Official Review · Reviewer_Z7w9 · 2025-11-01

**Soundness:** 3
**Presentation:** 3
**Contribution:** 3
**Rating:** 4
**Confidence:** 3

**Summary:**

This paper introduces Prompt-Agnostic Evolution (PAE), a framework designed to stabilize and accelerate training in Visual Prompt Tuning (VPT) for Vision Transformers. The authors identify that existing VPT variants suffer from unstable gradients, including shallow-layer stagnation and deep-layer oscillations. To address this, they propose two key components: 1) Modal Pre-Alignment (MPA): A frequency-domain initialization that aligns prompts with task-relevant frequency “shortcuts.” 2) Koopman-Lyapunov Discrete (KLD) system: A shared dynamical model where prompts evolve across layers under a Koopman operator with Lyapunov-based regularization for stability.
Experiments on FGVC and VTAB-1k benchmarks show 1–3% accuracy improvements and 1.5× faster convergence in terms of the number of required epochs.

**Strengths:**

1. Novel formulation: Reframing prompt tuning as a dynamical system using Koopman theory and Lyapunov stability is novel and mathematically grounded.

2. Comprehensive analysis: The paper clearly diagnoses VPT training instability through layer-wise gradient visualizations and supports it with quantitative results.

3. Strong empirical performance: PAE consistently improves various VPT baselines, showing strong generalization across tasks and benchmarks.

4. Prompt-agnostic applicability: The method is modular, adding no inference-time overhead and requiring no backbone modification.

5. Clear ablations: Ablation studies demonstrate the complementary roles of MPA and KLD, and verify robustness against random initialization and batch selection.

**Weaknesses:**

- Motivation clarity: While the dynamical-system framing is novel and interesting, the necessity of such complexity for solving gradient oscillation may be overstated. Simpler temporal regularization could have been compared. For example, could simpler smoothing (e.g., temporal moving average across layers) achieve comparable stability?

- Dependence on frequency bias: MPA relies on identifying “frequency shortcuts,” which may not exist or be stable in non-natural image domains, limiting transferability.

- Missing comparison on across-layer effects: The paper does not cite prior work such as [ref1], which examines how prompts interact across layers. A more direct comparison and analysis between that study’s findings and the proposed PAE framework would strengthen the discussion.

[ref1] Improving Visual Prompt Tuning for Self-supervised Vision Transformers, ICML 2023

- Questionable real-world efficiency: Although Fig. 1(a) and Table 1 show faster convergence in terms of epochs, the overall efficiency claim may be overstated. When accounting for MPA initialization overhead and the extra hyperparameters (α, β, K, w, r) that expand the tuning space, the total wall-clock time might actually increase. Hence, the practical benefit in large-scale or hyperparameter-sensitive scenarios remains uncertain.

**Questions:**

Please see above

---

> ### Author Response · Authors · 2025-11-21
> **Response to Reviewer Z7w9 (1/2)**
>
> W1: Motivation Clarification and Comparison with Simpler Regularization.
>
> We appreciate this comment. While simple temporal smoothing (e.g., EMA) can reduce noise, our analysis shows that it is insufficient for achieving stable gradient evolution and effective task-directional prompt optimization.
>
> 1. **Empirical comparison with smoothing:** We implemented an exponential moving average baseline (“VPT+EMA”, shown in Fig. 1). EMA does smooth raw VPT gradients and reduces some severe oscillations (Fig. 1c). However, both shallow and deep gradients quickly collapse to a very low magnitude around $10^{-2}$–$10^{-3}$, far below the stable $10^{-1}$ band maintained by VPT+$\mathtt{PAE}$. Correspondingly, in Fig. 1(a), VPT+EMA reaches $90.14%$ top-1 at epoch 93, compared to $90.77%$ at epoch 78 for VPT, while VPT+$\mathtt{PAE}$ attains $92.27%$ at epoch 32. Thus, although EMA slightly improves stability over vanilla VPT, both the stability of the gradient dynamics and the final performance remain **clearly inferior** to our VPT+$\mathtt{PAE}$.
> 2. **Limitations of Smoothing**: A moving average acts as a scalar temporal filter: all dimensions are smoothed equally, with no mechanism to distinguish useful long-range dynamics from noisy components. This prevents it from shaping layer-wise evolution in a way that preserves informative modes.
> 3. **Why $\mathtt{PAE}$ is fundamentally more expressive:** $\mathtt{PAE}$ learns a matrix operator $\mathbf{K}$. Our spectral analysis (Fig. 10) shows this operator maintains eigenvalues inside the unit disk (ensuring stability) while selectively retaining informative modes near the real axis. This enables stable yet discriminative evolution across layers which is something scalar smoothing cannot achieve.
> 4. **Note** that scalar smoothing is actually a **degenerate** Koopman operator with $\mathbf{K}=λI$, i.e., all eigenvalues are identical, so it cannot realize the non-trivial modal structure and selective stability that our learned operator exhibits in Fig. 10.
>
>
> W2: Dependence on frequency bias.
>
> Frequency patterns are not unique to natural images.
>
> 1. **Evidence:** Our method works by identifying discriminative spectral energy, which exists in any structured data (e.g., VTAB-1k benchmark includes cell boundaries in pathology, road networks in satellite imagery).
> 2. **Results:** We validated this on **VTAB-1k**, which explicitly includes non-natural tasks (Medical, Satellite, Synthetic). As shown in Table 3, VPT+MPA consistently outperforms the baseline on the **Specialized** and **Structured** groups, confirming that frequency shortcuts are effective across domains.
>
>
> W3: Missing comparison on across-layer effects
>
> We have added a direct comparison with GatePT [1], which models across-layer interaction explicitly.
>
> 1. In the revised paper, we employ **Centered Kernel Alignment (CKA)** [2] for this analysis because it is a robust, rotation-invariant metric specifically designed to measure the representational similarity between different neural network layers. This allows us to diagnose the *dynamics* of the learned prompts: assessing whether they are redundant (high global similarity) or evolving progressively (diagonal similarity).
> 2. **CKA Analysis (Fig. 7):** Figure 7 visualizes layer-wise prompt similarity. GatePT [1] introduces heuristic gating that mildly smooths cross-layer correlations; however, it still produces globally entangled prompts with uniformly high similarity across almost all layers. In contrast, VPT+$\mathtt{PAE}$ exhibits a distinct diagonal structure, where similarity is highest locally and decays as layers become farther apart. This pattern indicates a progressive, depth-aware evolution of the prompt state which is precisely the behavior induced by our learned Koopman operator and stability regularization.
>
> [1] Improving Visual Prompt Tuning for Self-supervised Vision Transformers, ICML 2023.
> [2] Similarity of neural network representations revisited, ICML 2019.

---

> > ### Author Response · Authors · 2025-11-21
> > **Response to Reviewer Z7w9 (2/2)**
> >
> > W4: Questionable real-world efficiency.
> >
> > We agree that "epochs" can be a proxy. Here is the breakdown of actual wall-clock efficiency:
> >
> > | Backbone | Tuned/ Total   | Ratio (%)       | MPA Init. Time | Training Time (per Epoch) | MPA Overhead (vs. 100 Epochs) | Equivalent Training Epochs |
> > | -------- | -------------- | --------------- | -------------- | ------------------------- | ----------------------------- | -------------------------- |
> > | ViT-B/16 | 0.65M / 86.6M  | $\approx$ 0.75% | $\approx$ 74s  | $\approx$ 14s             | $\approx$ 5.3%                | $\approx$ 5.3              |
> > | ViT-L/16 | 1.25M / 304.6M | $\approx$ 0.41% | $\approx$ 373s | $\approx$ 42s             | $\approx$ 8.0%                | $\approx$ 8.9              |
> > | ViT-H/14 | 1.73M / 632.5M | $\approx$ 0.27% | $\approx$ 765s | $\approx$ 105s            | $\approx$ 7.0%                | $\approx$ 7.3              |
> >
> > 1. **Wall-clock Efficiency:** The one-time MPA initialization takes only ~74s (ViT-B), equivalent to roughly **5.3 training epochs**. By paying this small cost, we accelerate convergence by **1.78x** (saving ~40-60 epochs). Even after accounting for the initialization time, the net efficiency gain remains substantial. Combined with the fact that $\mathtt{PAE}$ introduces zero inference overhead, the overall efficiency improvement is significant.
> >
> > 2. **Hyperparameters Sensitivity and Design Rationale:** Our sensitivity studies show that all key hyperparameters have broad stable regions. As seen in **Table 6**, the frequency parameters $(w, r)$ vary accuracy by <0.8% across most settings; **Table 7** shows similar stability for the loss weights $(\alpha, \beta)$; and **Fig. 9** demonstrates a clear performance plateau for the Koopman dimension $K \in [128,320]$. These results collectively indicate that the method is not brittle. Based on our emperical results, we provide detailed design rationales for each hyperparameter:
> >
> >    • **Frequency parameters $(w, r)$.**
> >    MPA focuses on low–mid frequency components because prior analysis [1] shows this band consistently captures shortcut-prone, discriminative structure across architectures. We therefore restrict $(w, r)$ to this theoretically motivated region and run a small sweep within it. Table 6 shows performance is stable across this entire band.
> >    **Suggested range:** $(w=16-24, r=8-12)$.
> >
> >    • **Loss weights $(\alpha, \beta)$.**
> >    These weights balance the constraint and regularization terms so that their gradient magnitudes are comparable to the base classification loss, following standard practice for stable constrained optimization. Table 7 confirms robustness across a wide interval.
> >    **Suggested range:** $(\alpha=0.5-0.7, \beta=0.2)$.
> >
> >    • **Koopman dimension $K$.**
> >    \(K\) controls the capacity of the learned operator. Once sufficient capacity is reached, performance should saturate—which Fig. 9 shows clearly for $K\in[128, 320]$. We therefore select a midpoint and use it consistently across datasets and architectures.
> >    **Suggested value:** \($K$ = 256\).
> >
> >    Overall, each hyperparameter has a clear underlying motivation (spectral selection, gradient balance, model capacity), and all exhibit broad, empirically validated stability ranges.
> >
> >    [1] Do ImageNet-trained models learn shortcuts? The impact of frequency shortcuts on generalization. CVPR 2025.

---

### Author Response · Authors · 2025-11-21
**General Response**

We sincerely thank you for your constructive and insightful comments. We have carefully revised our paper to address your concerns. All significant changes and new experiments in the revised PDF are **highlighted**.

**Summary of Major Updates:**

1. **Expanded Scope & Architectures:** We added experiments on **ViT-L/16, ViT-H/14** (Large/Huge models), **Swin-B** (Hierarchical), **MAE** (Self-supervised), and **SETR** (Segmentation architecture) to demonstrate scalability and robustness (**Table 2, Fig. 6, and Fig. 7**).
2. **New Task (Dense Prediction):** We added **Semantic Segmentation** results on **ADE20K** to demonstrate generalization beyond classification (**Table 2**).
3. **Stronger Baselines:** We added comparisons with **LoRA, SPT-LoRA, and DM-LoRA** in the Appendix (**Tables 8-11**).
4. **In-depth Analysis:** We included **CKA (Fig. 7)**, **Intra-class variance (Fig. 8)** and **Spectral analysis (Fig. 10)** to visualize the mechanism of our dynamical system.
5. **Clarifications:** We revised the **Introduction** to discuss projection-based methods and the **Results** to quantify initialization costs.

We believe these updates significantly strengthen the paper and fully address the concerns regarding generalization, efficiency, and mechanism.

---

### Meta-Review · Area_Chair_H12f · 2026-01-06

**Summary:**

Reviewer PEzU expressed concerns regarding the experimental scope and the theoretical assumptions of the KLD framework and this was addressed by the authors and acknowledged by the reviewer

Reviewer 9KV8 requested clarification on differences from projection-based prompt tuning methods and clearer motivation for the frequency shortcuts approach which was addressed by the authors and acknowledged by the reviewer

Reviewer oUNy raised concerns about hyperparameter sensitivity limiting plug-and-play usability, and questioned whether the linear operator assumption limits expressiveness for complex tasks or deep networks. These issues were addressed by the authors and acknowledged by the reviewer.

**Reviewer Concerns:**

Three reviwers oUNy, 9KV8 and PEzU acknowledged the rebuttal addressed their issues and moved their scores up. Z7w9 also had the hyper paramter concern which was addressed by the rebuttal.

**Reviewer Scores:**

PEzU raised their score to  a 8
9KV8 kept their score at 8
oUNy stated they are going to increase their score and we can assume that its a 5/6
Z7w9 had a partial address in their issues so we can expect a 5/6 there too.

---

### Decision · Program_Chairs · 2026-01-26

Accept (Poster)